# DATA SHARING WITHOUT REWARDS IN MULTI-TASK OFFLINE REINFORCEMENT LEARNING

## ABSTRACT

Offline reinforcement learning (RL) bears the promise to learn effective control policies from static datasets but is thus far unable to learn from large databases of heterogeneous experience. The multi-task version of offline RL enables the possibility of learning a single policy that can tackle multiple tasks and allows the algorithm to share offline data across tasks. Recent works indicate that sharing data between tasks can be highly beneficial in multi-task learning. However, these benefits come at a cost – for data to be shared between tasks, each transition must be annotated with reward labels corresponding to other tasks. This is particularly expensive and unscalable, since the manual effort in annotating reward grows quadratically with the number of tasks. Can we retain the benefits of data sharing without requiring reward relabeling for every task pair? In this paper, we show that, perhaps surprisingly, under a binary-reward assumption, simply utilizing data from other tasks with constant reward labels can not only provide substantial improvement over only using the single-task data and previously proposed success classifiers, but it can also reach comparable performance to baselines that take advantage of the oracle multi-task reward information. We also show that this performance can be further improved by selectively deciding which transitions to share, again without introducing any additional models or classifiers. We discuss how these approaches relate to each other and baseline strategies under various assumptions on the dataset. Our empirical results show that it leads to improved performance across a range of different multi-task offline RL scenarios, including robotic manipulation from visual inputs and ant-maze navigation.

## 1 INTRODUCTION

Offline reinforcement learning (RL) provides the promise of a fully data-driven framework for learning performant policies. To avoid costly active data collection and exploration, offline RL methods utilize a previously collected dataset to extract the best possible behavior, making it feasible to use RL to solve real-world problems where active exploration is expensive, dangerous, or otherwise infeasible (Zhan et al., 2021; de Lima & Krohling, 2021; Wang et al., 2018; Kalashnikov et al., 2018). However, this concept is only viable when a significant amount of data for the target task is available in advance. A more realistic scenario might allow for a much smaller amount of task-specific data, combined with a large amount of task-agnostic data, that is not labeled with task rewards and some of which may not be relevant. For example, if our goal is to train a robot to perform a new manipulation task (e.g., cutting an onion), we might have some data of the robot (suboptimally) attempting that task, perhaps collected under human teleoperation and manually labeled with rewards, combined with plentiful data of other tasks, some of which might be structurally related (e.g., picking up an onion, or cutting a carrot). This scenario presents several questions: How do we decide which prior data should be included when learning the new task? And how do we determine which reward labels to use for this prior data?

Prior methods have offered several potential answers to these two questions, typically in isolation. For the first question, it has been recently observed that a naïve sharing strategy of sharing data from all tasks can be highly suboptimal (Kalashnikov et al., 2021), and some works have proposed both manual (Kalashnikov et al., 2021) and automated (Yu et al., 2021a; Eysenbach et al., 2020) data-sharing strategies that prioritize the most structurally similar prior data. Most such methods assume that this shared data can be automatically relabeled with the reward function for the new task (Kalashnikov et al., 2021; Yu et al., 2021a; Eysenbach et al., 2020), but the assumption that we have access to the

functional form of this reward is a strong one: for example, in many real-world settings, the reward might require human labeling or human-provided examples (Cabi et al., 2019; Finn et al., 2016b). To this end, some prior works have proposed learning classifiers for reward labeling (Fu et al., 2018b; Xie et al., 2018; Singh et al., 2019), or other automated mechanisms (Konyushkova et al., 2020). But these mechanisms themselves add complexity and potential brittleness to the pipeline. Thus, we aim to devise a simple unified method that determines which data to share and which rewards to use, with minimal supervision and no additional modeling and learning.

In this paper, we make the potentially surprising observation that data from other tasks can be utilized with naïve constant reward labels, when the MDP consists of binary rewards. We show that this simple method, which does not involve learning any additional models or classifiers, can outperform more sophisticated techniques in practice. Our approach simply utilizes data from other tasks with a constant reward label (e.g., $r = 0$), and uses a value-aware strategy to decide which prior transitions to include for the new task. This strategy, based on the conservative data sharing (CDS) technique proposed in prior work (which assumes oracle reward access) (Yu et al., 2021a), also does not require learning any additional model and simply uses the Q-function that is already learned as part of the RL process.

Our main contribution, which we call conservative unsupervised data sharing (CUDS), is a technique for sharing data in multi-task offline RL that does not require any reward labels or reward function access for the task-agnostic data, and requires no additional model or classifier. To achieve that, our method assumes a particular form of the MDP that consists of binary rewards. We discuss the behaviors of our methods, showing that, even without ground truth reward labels, our simple data sharing scheme achieves Q-values that are lower-bounded by the Q-values obtained with sharing all data with the ground-truth rewards and can be combined CDS to selectively filter out potentially irrelevant data under different assumptions on the structure of the dataset. Our empirical evaluation conducted over various multi-task offline RL scenarios such as robotic manipulation from visual inputs and ant-maze navigation shows that this approach improves over the performance of more sophisticated techniques that either learn the reward function explicitly, or utilize other methods to propagate reward labels. In addition, we show that the proposed approach is comparable to an oracle baseline that has access to true multi-task rewards.

## 2 RELATED WORK

**Offline RL.** Offline RL (Ernst et al., 2005; Riedmiller, 2005; Lange et al., 2012; Levine et al., 2020) considers the problem of learning a policy from a static dataset without interacting with the environment, which has shown promises in many practical applications such as robotic control (Kalashnikov et al., 2018; Mandlekar et al., 2020; Rafailov et al., 2021), NLP (Jaques et al., 2019), healthcare (Shortreed et al., 2011; Wang et al., 2018), education (de Lima & Krohling, 2021), electricity supply (Zhan et al., 2021) and UI design (Apostolopoulos et al., 2021). The main challenge of offline RL is the distributional shift between the learned policy and the behavior policy (Fujimoto et al., 2018; Kumar et al., 2019), which can cause erroneous value backups due to out-of-distribution actions generated by the learned policy. To address this issue, prior methods have constrained the learned policy to not deviate much from the behavior policy via policy regularization (Liu et al., 2020; Jaques et al., 2019; Wu et al., 2019; Zhou et al., 2020; Kumar et al., 2019; Siegel et al., 2020; Peng et al., 2019; Zhou et al., 2020; Kostrikov et al., 2021; Ghasemipour et al., 2021), conservative value functions (Kumar et al., 2020; Sinha & Garg, 2021), an auxiliary behavioral cloning loss (Fujimoto & Gu, 2021) and model-based training with conservative penalties (Yu et al., 2020d; Kidambi et al., 2020; Argenson & Dulac-Arnold, 2020; Swazinna et al., 2020; Matsushima et al., 2020; Lee et al., 2021; Yu et al., 2021b).

**Multi-Task RL and data sharing.** Multi-task RL (Wilson et al., 2007; Parisotto et al., 2015; Teh et al., 2017; Espeholt et al., 2018; Hessel et al., 2019; Yu et al., 2020a; Xu et al., 2020; Yang et al., 2020; Kalashnikov et al., 2021; Sodhani et al., 2021; Stooke et al., 2021) enables the goal of learning a single policy that solves multiple skills efficiently. Despite the promising results, multi-task RL suffers from three main challenges, optimization difficulties (Schaul et al., 2019; Hessel et al., 2019; Yu et al., 2020a), effective weight sharing for learning shared representations (Parisotto et al., 2015; Teh et al., 2017; Espeholt et al., 2018; Xu et al., 2020; D'Eramo et al., 2019; Sodhani et al., 2021; Stooke et al., 2021), and sharing data across different tasks (Eysenbach et al., 2020; Kalashnikov et al., 2021; Yu et al., 2021a). We consider the multi-task offline RL setting and focus

on the challenge of sharing data across different tasks. Prior works share data across tasks based on metrics such as learned Q-values (Eysenbach et al., 2020; Li et al., 2020; Yu et al., 2021a), human domain knowledge (Kalashnikov et al., 2021), the distance to the target goals in goal-conditioned settings (Andrychowicz et al., 2017; Pong et al., 2018; Nair et al., 2018; Liu et al., 2019; Sun et al., 2019; Lin et al., 2019; Huang et al., 2019; Lynch & Sermanet, 2020; Yang et al., 2021; Chebotar et al., 2021), and the learned distance with robust inference in the offline meta-RL setting (Li et al., 2019). However, all of these either require access to the functional form of the reward functions of each task in order to relabel the rewards or are limited to goal-conditioned settings. Therefore, they are not applicable to the multi-task offline RL problem that we consider where only the reward label of the originally-executed task is provided. Our work addresses this issue via simply relabeling the data shared from other tasks with a constant value and uses the conservative data sharing strategy (Yu et al., 2021a) to further improve the performance.

**RL with unlabeled data.** Prior works tackle the problem of learning from data without reward labels via either directly imitating expert trajectories (Pomerleau, 1988; Ross & Bagnell, 2012; Ho & Ermon, 2016), learning reward functions from expert data using inverse RL (Abbeel & Ng, 2004; Ng & Russell, 2000; Ziebart et al., 2008; Finn et al., 2016a; Fu et al., 2018a;b; Konyushkova et al., 2020), or learning a reward / value classifier that discriminates successes and failures (Xie et al., 2018; Singh et al., 2019; Eysenbach et al., 2021). These algorithms require online data collection and do not consider the offline RL setting. Singh et al. (2020) considers the single-task offline setting with both task-specific datasets and task-agnostic prior datasets and relabel the unlabeled prior data as failures since these prior transitions cannot solve the task. Our method is not limited to such single-task settings and instead considers the more general multi-task offline RL with data-sharing problem.

## 3 PRELIMINARIES

**Multi-task RL.** Standard multi-task RL considers a multi-task Markov decision process (MDP), $\mathcal{M} = (\mathcal{S}, \mathcal{A}, P, \gamma, \{R_i, i\}_{i=1}^N)$, where $\mathcal{S}$ and $\mathcal{A}$ denote the state and action spaces respectively, $P(\mathbf{s}'|\mathbf{s}, \mathbf{a})$ denotes the dynamics, $\gamma \in [0, 1)$ is the discount factor, and $R_1, \cdots, R_N$ correspond to reward functions of different tasks $i \in [N]$ for total number of $N$ tasks where $[N]$ is the shorthand for $\{1, 2, \ldots, N\}$. In our setting, we assume a binary per-task $R_i \in \{0, 1\}$, where 1 denotes success of the task and 0 otherwise. Note that the dynamics are assumed to be the same across all tasks, which is not entirely general but is indeed practical in many problem settings as noted in Yu et al. (2021a) and stands as a common assumption in prior data sharing works (Yu et al., 2021a; Kalashnikov et al., 2021; Eysenbach et al., 2020). Regardless, there are many practical scenarios with changing rewards and invariant dynamics such as various object manipulation objectives (Xie et al., 2018), different goal navigation tasks (Fu et al., 2020), and distinct user preferences (Christiano et al., 2017). The goal of multi-task RL is to find a task-conditioned policy $\pi(\mathbf{a}|\mathbf{s}, i)$ that expected return in a multi-task MDP: $\pi^*(\mathbf{a}|\mathbf{s}, \cdot) := \arg\max_\pi \mathbb{E}_{i \sim [N]} \mathbb{E}_{\pi(\cdot|\cdot, i)}[\sum_t \gamma^t R_i(\mathbf{s}_t, \mathbf{a}_t)]$. Note that it is possible to model the policies for each task independently as $\{\pi_1(\mathbf{a}|\mathbf{s}), \cdots, \pi_N(\mathbf{a}|\mathbf{s})\}$ without any weight sharing. In our work, we use the single task-conditioned policy to study data sharing and do not consider the weight sharing aspect, which is orthogonal to the focus of the paper, which is also noted in Yu et al. (2021a).

**Multi-task offline RL and data sharing.** Multi-task offline RL considers the problem of learning the multi-task policy $\pi(\mathbf{a}|\mathbf{s}, i)$ from a static multi-task dataset with $\mathcal{D} = \cup_{i=1}^N \mathcal{D}_i$ where $\mathcal{D}_\rangle = \{(\mathbf{s}_j, \mathbf{a}_j, \mathbf{s}'_j, r_j)\}_{j=1}^M$ is the per-task dataset. $\mathcal{D}_i$ is generated by a behavior policy $\pi_\beta(\mathbf{a}|\mathbf{s})$, without any interaction with the environment. The most straightforward approach to learn $\pi(\mathbf{a}|\mathbf{s}, i)$ would be train it for task $i$ only using $\mathcal{D}_i$. However, sharing data from different tasks to task $i$ has been shown to be conducive in the multi-task offline RL setting (Kalashnikov et al., 2021; Yu et al., 2021a). To do so, prior works (Eysenbach et al., 2021; Kalashnikov et al., 2021; Yu et al., 2021a) assume access to the functional form of the reward $r_i$, which is a rather strong assumption that is usually impractical to specify in practical applications due to the challenge of reward specification. The next straightforward approach is to naïvely sharing data across all tasks, denoted as Sharing All. Formally, Sharing All defines the dataset of transitions relabeled from task $j$ to task $i$ as $\mathcal{D}_{j \to i}$ and the method can be then defined as $\mathcal{D}_i^{\text{eff}} := \mathcal{D}_i \cup (\cup_{j \neq i} \mathcal{D}_{j \to i})$, where $\mathcal{D}_i^{\text{eff}}$ denotes the effective dataset for task $i$. While Sharing All improves over not sharing data, as shown in Yu et al. (2021a), Sharing All leads to distributional shift that could degrade performance in certain situations (Yu et al., 2021a). In our work, we focus on the CDS (Yu et al., 2021a), which relabels data that aims to mitigate the

distributional shift introduced by sharing other task data. CDS addresses such an issue by proposing a conservative data sharing strategy as follows:

$$\mathcal{D}_i^{\text{eff}} = \mathcal{D}_i \cup (\cup_{j \neq i}\{(\mathbf{s}_j, \mathbf{a}_j, \mathbf{s}_j', r_i) \in \mathcal{D}_{j \to i} : \Delta^\pi(\mathbf{s}, \mathbf{a}) \geq 0\}), \tag{1}$$

where $\mathbf{s}_j, \mathbf{a}_j, \mathbf{s}_j'$ denote the transition from $\mathcal{D}_j$, $r_i$ denotes the reward of $\mathbf{s}_j, \mathbf{a}_j, \mathbf{s}_j'$ relabeled for task $i$, $\pi$ denotes the task-conditioned policy $\pi(\cdot|\cdot, i)$, $\Delta^\pi(\mathbf{s}_j, \mathbf{a}_j)$ is the condition that shares data only if the expected Q-value of the relabeled transition exceeds the top $k$-percentile of the Q-values of the original task data, i.e.

$$\Delta^\pi(\mathbf{s}, \mathbf{a}) := \hat{Q}^\pi(\mathbf{s}, \mathbf{a}, i) - P_{k\%}\left\{\hat{Q}^\pi(\mathbf{s}', \mathbf{a}', i) : \mathbf{s}', \mathbf{a}' \sim \mathcal{D}_i\right\}. \tag{2}$$

Beyond controlling the distributional shift introduced in data sharing, multi-task offline RL also needs to address the main challenge in standard offline RL, which is the distributional shift between the learned policy $\pi$ and the behavior policy $\pi_\beta$. To handle both types of distributional shifts, CDS (Yu et al., 2021a) combines the conservative data sharing and the constrained policy optimization problem and arrives at the following objective:

$$\forall i \in [N], \ \pi^*(\mathbf{a}|\mathbf{s}, i) := \arg\max_\pi \ J_{\mathcal{D}_i^{\text{eff}}}(\pi) - \alpha D(\pi, \pi_\beta^{\text{eff}}), \tag{3}$$

where $\pi_\beta^{\text{eff}}(\mathbf{a}|\mathbf{s}, i)$ is the effective behavior policy for task $i$ denoted as $\pi_\beta^{\text{eff}}(\mathbf{a}|\mathbf{s}, i) := |\mathcal{D}_i^{\text{eff}}(\mathbf{s}, \mathbf{a})|/|\mathcal{D}_i^{\text{eff}}(\mathbf{s})|$, $J_{\mathcal{D}_i^{\text{eff}}}(\pi)$ denotes the average return of policy $\pi$ in the empirical MDP induced by the effective dataset, and $D(\pi, \pi_\beta^{\text{eff}})$ denotes a divergence measure (e.g., KL-divergence (Jaques et al., 2019; Wu et al., 2019), fisher divergence (Kostrikov et al., 2021), MMD distance (Kumar et al., 2019) or $D_{\text{CQL}}$ from conservative Q-learning (Kumar et al., 2020)) between the learned policy $\pi$ and the effective behavior policy $\pi_\beta^{\text{eff}}$. While optimizing Eq. 3 with Eq. 1 as the data sharing scheme is able to mitigate distributional shift and improve over multi-task offline RL without sharing data and naïvely sharing data across all tasks as shown in Yu et al. (2021a), it requires the assumption of the access to the functional form of the reward functions, which is rather strong and make application of data sharing to real-world applications impractical. We will instead present a simple yet effective data sharing and relabeling scheme in the setting where we do not make such an assumption and instead, only have the reward labels for originally commanded task in the following section.

## 4 DATA SHARING WITHOUT REWARDS IN MULTI-TASK OFFLINE RL

The goal of our method is to enable effective data sharing across different tasks without access to the functional form of the reward functions for each task. Data from each task is only labeled for that particular task, and we do not know a priori which data is relevant to each task. Effective data sharing therefore requires resolving two questions: (i) which data from other tasks should we use for a given task? and (ii) how do we label this data with rewards? One simple approach is to annotate all available data from other tasks with some "proxy" reward signal, and treat it no differently from data that is already labeled. That is, after relabeling with the proxy reward, we can simply put these transitions into the replay buffer of a value-based offline RL method. But how can we obtain a reliable proxy reward signal? Next, we will discuss two variants of our method in Section 4.1, understanding of both variants in Section 4.2, and practical implementations in Section 4.3.

### 4.1 CONSERVATIVE UNSUPERVISED DATA SHARING

Prior work assumes that it is necessary to relabel prior data with some estimate of the true reward function so that the proxy reward closely reflects the true reward. We take a different approach, and instead argue that, under some assumptions, we can obtain many of the benefits of data sharing simply by labeling the multi-task data with the lowest possible reward, which we assume to be 0 without loss of generality in the binary-reward setting. We refer to this simple strategy as unsupervised data sharing (UDS). Naïve UDS prescribes sharing data from every task to every other task, and labels the shared data with a reward value of 0. Formally, for each task $i \in [N]$, we define the UDS procedure as follows:

$$\mathcal{D}_i^{\text{eff}} = \mathcal{D}_i \cup \{(\mathbf{s}_j, \mathbf{a}_j, \mathbf{s}_j', 0) \in \mathcal{D}_{j \to i} : \forall j \in [N] \setminus \{i\}\}. \tag{4}$$

Intuitively, UDS relabels data shared from other tasks with the lowest possible reward, hence making the learned Q-functions more conservative that the data sharing scheme with the oracle rewards. We will show that UDS learns Q-values that are lower-bounded by the Q-values learned by the naïve Sharing All scheme with true reward relabeling, and can be information-theoretically optimal in offline RL thanks to such conservatism. Our empirical results in Section 5 also suggests that the benefits from data sharing outweigh the downsides of reward bias in practice. Next, we move on from the choice of proxy reward, to the choice of which data should be shared.

While UDS is simple yet effective in multi-task data sharing without reward relabing, naïvely sharing data from all other tasks with zero rewards in the offline RL setting can result in overly conservative Q-functions and policies. To further refine this strategy, we can adapt the CDS algorithm (Yu et al., 2021a) detailed in Section 3 to filter out irrelevant transitions from other tasks, and only share those transitions that are likely to be informative. We call this strategy conservative unsupervised data sharing (CUDS). We define the CUDS strategy as follows:

$$\mathcal{D}_i^{\text{eff}} = \mathcal{D}_i \cup \{(\mathbf{s}_j, \mathbf{a}_j, \mathbf{s}_j', 0) \in \mathcal{D}_{j \to i} : \Delta^\pi(\mathbf{s}, \mathbf{a}) \geq 0 \ \ \forall j \in [N] \setminus \{i\}\}. \tag{5}$$

As we will discuss in the next subsection, CUDS is able to select potentially useful transitions under certain structural assumptions on the multi-task offline dataset, and therefore produce Q-values that are not as excessively conservative as those produced by UDS. As shown in Section 5, our empirical evaluation further validates that CUDS improves over UDS and prior approaches.

---

**Algorithm 1** (Conservative) Unsupervised Data Sharing

---

**Require:** Multi-task offline datasets $\cup_{i=1}^N \mathcal{D}_i$.
1: Randomly initialize policy $\pi_\theta(\mathbf{a}|\mathbf{s}, i)$.
2: **for** $k = 1, 2, 3, \cdots,$ **do**
3:     Initialize $\mathcal{D}^{\text{eff}} \leftarrow \{\}$
4:     **for** $i = 1, \cdots, N$ **do**
5:         $\mathcal{D}_i^{\text{eff}} = \mathcal{D}_i \cup \{(\mathbf{s}_j, \mathbf{a}_j, \mathbf{s}_j', 0) \in \mathcal{D}_{j \to i} \ \forall j \in [N] \setminus \{i\}\}$ (UDS) or $\mathcal{D}_i^{\text{eff}} = \mathcal{D}_i \cup \{(\mathbf{s}_j, \mathbf{a}_j, \mathbf{s}_j', 0) \in \mathcal{D}_{j \to i} : \Delta^\pi(\mathbf{s}, \mathbf{a}) \geq 0 \ \forall j \in [N] \setminus \{i\}\}$ using Eq. 2 (CUDS).
6:     Perform policy improvement by solving Eq. 3 by sampling data from $\mathcal{D}^{\text{eff}}$.

---

## 4.2 Understanding the Behavior of UDS and CUDS

In this section, we aim to understand the behavior of the UDS and CUDS. We first consider the UDS scheme, which simply shares all available data from other tasks, and labels the reward for each transition from other tasks as $0$. When instantiated with CQL as the offline RL method, the Q-values of a given policy learned by UDS for each task $i$ are the fixed point of the recursion:

$$\widehat{Q}^{k+1}(\mathbf{s}, \mathbf{a}, i) \leftarrow \widehat{r}(\mathbf{s}, \mathbf{a}, i) + \gamma \mathbb{E}_{\mathbf{s}' \sim \widehat{P}(\mathbf{s}'|\mathbf{s}, \mathbf{a}), \pi(\mathbf{a}'|\mathbf{s}', i)} \left[ \widehat{Q}^k(\mathbf{s}', \mathbf{a}', i) \right] - \alpha \left( \frac{\pi(\mathbf{a}|\mathbf{s}, i)}{\widehat{\pi}_\beta(\mathbf{a}|\mathbf{s}, i)} - 1 \right), \tag{6}$$

where $\widehat{r}(\mathbf{s}, \mathbf{a}, i) = 0$ for all $(\mathbf{s}, \mathbf{a}) \in \mathcal{D}_{j \to i}.j \neq i$, and $\widehat{r}(\mathbf{s}, \mathbf{a}, i)$ is equivalent to the empirical reward observed otherwise (Kumar et al., 2020). We will now try to understand how UDS compares to the No Sharing strategy, which only uses the labeled data for training. Note that this comparison is non-trivial since, while UDS utilizes a larger dataset, it can induce significant reward bias during training. However, by assumption, $0$ is the lowest possible reward, we would intuitively expect that UDS should be *more* conservative, compared to Sharing All that relabels with the true reward. While we may surmise that being too conservative on unlabeled data may be suboptimal, conservatism has been shown to be information-theoretically optimal (Jin et al., 2021; Rashidinejad et al., 2021) in offline RL and bandit problems. Even though an unlabeled dataset provides us with information about environment dynamics, it does not provide information about rewards, and any optimistic estimate of reward on this data may lead to poor performance in the worst case.

We formally derive the performance guarantee for UDS in Proposition F.1 using the framework of safe-policy improvement and discuss cases where it can perform better than No Sharing. We discuss in Appendix F.2.3 that UDS can perform better than No Sharing in long horizon tasks as well as in cases where the unlabeled dataset consists of similar proportions of various state-action pairs as the labeled dataset. Please refer to this section for the theoretical results. Our bounds utilize a new

technique that allows us to prove tighther, non-trivial bounds for UDS. despite the pessimism which is discussed in Appendix F.2.1.

To understand the behavior of CUDS, we will consider a simple abstract model of CUDS-style relabeling. In the tabular setting, this model updates the Q-function to match the target (conservative) Q-value if the transition is selected for the update, and retains the old table entry otherwise. Formally, consider a binary vector $\mathbf{w} \in \mathbb{R}^{|\mathcal{S}| \times |\mathcal{A}|}$ that indicates whether a corresponding state-action pair $(\mathbf{s}, \mathbf{a})$ is utilized for the backup or not. Then, our weighted scheme performs the following backups:

$$\widehat{Q}^{k+1}(\mathbf{s}, \mathbf{a}, i) = \mathbf{w}(\mathbf{s}, \mathbf{a}) \left[ \widehat{r}(\mathbf{s}, \mathbf{a}, i) - \alpha \left( \frac{\pi(\mathbf{a}|\mathbf{s}, i)}{\hat{\pi}_\beta(\mathbf{a}|\mathbf{s}, i)} - 1 \right) + \gamma \mathbb{E}_{\mathbf{s}', \mathbf{a}' \sim \widehat{P}(\cdot|\mathbf{s}, \mathbf{a}), \pi(\cdot|\mathbf{s}', i)} \left[ \widehat{Q}^k(\mathbf{s}', \mathbf{a}', i) \right] \right]$$

$$+ (1 - \mathbf{w}(\mathbf{s}, \mathbf{a}))\widehat{Q}^k(\mathbf{s}, \mathbf{a}, i).$$

(7)

Equation 7 can be intuitively understood as performing a conservative backup from the actual transition observed in the dataset when the binary weight $\mathbf{w}(\mathbf{s}, \mathbf{a}) = 1$, and simply truncating the Bellman backup and retaining the previous Q-values $\widehat{Q}^k(\mathbf{s}, \mathbf{a}, i)$, otherwise. For example, CUDS performs a conservative backup with $\widehat{r}(\mathbf{s}, \mathbf{a}, i) \leq \widehat{r}_{\text{Sharing All}}(\mathbf{s}, \mathbf{a}, i)$ only on transitions where $\mathbf{w}(\mathbf{s}, \mathbf{a}) = \mathbb{I}[\Delta^\pi(\mathbf{s}, \mathbf{a}) \geq 0]$.

To understand how this affects the resulting Q-function, we consider two structural conditions on the offline dataset: **(1)** a scenario where no trajectory in the relabeled dataset for a given target task $\mathcal{D}_{j \to i}$ actually visits state-action tuples that were observed in $\mathcal{D}_i$, and **(2)** when trajectories in $\mathcal{D}_{j \to i}$ overlap with at least a fraction of state-action tuples in the original labeled data for this task $\mathcal{D}_i$. We will abstract CUDS as utilizing $\mathbf{w}^k(\mathbf{s}, \mathbf{a}) = \mathbb{I}[\widehat{Q}^k(\mathbf{s}, \mathbf{a}, i) \geq \iota]$ for some threshold $\iota$ (see Eqn. 2).

**Remark 4.1** (CUDS reduces to no sharing under condition **(1)**). *When the trajectories in the unlabeled, relabeled dataset do not overlap with any trajectory in the labeled dataset for a given task, any backup performed by CUDS on an unlabeled transition will eventually drive its Q-value to $0$ as $k \to \infty$. Thus, CUDS weights $\mathbf{w}^k(\mathbf{s}, \mathbf{a})$ will eventually take on $0$ values for such transitions, and will not be selected by the future weights, i.e., $\mathbf{w}^j(\mathbf{s}, \mathbf{a}) = 0 \ \forall j \geq k + 1$.*

Perhaps unsurprisingly, when the unlabeled data has no overlap with the labeled data, CUDS reduces to no sharing. However, the more practically relevant case is when the unlabeled data overlaps with the labeled data. We consider the scenario when UDS has been run initially to obtain a starting set of Q-values, $\widehat{Q}^0(\mathbf{s}, \mathbf{a}, i)$, which defines the initial weight vector.

**Remark 4.2** (CUDS selects more useful unlabeled transitions). *Imagine a transitions $(\mathbf{s}, \mathbf{a}, \mathbf{s}', 0) \in \mathcal{D}_{j \to i}$ for which the next state (and the next policy action) $(\mathbf{s}', \mathbf{a}')$ are observed in the labeled dataset (denoted $\mathcal{D}_i$). This transition will will attain large initial Q-values $\widehat{Q}^0(\mathbf{s}, \mathbf{a}, i)$ if executing the policy after $(\mathbf{s}', \mathbf{a}')$ eventually reaches the state that corresponds to a high reward of 1.0, due to the Bellman backup component of CUDS. However, on the flip side, these backups performed by CUDS are conservative, and performing more backups can reduce the Q-value. Two scenarios might then arise: (i) the Q-values eventually decrease and CUDS is deactivated, i.e., $\exists k, \ \mathbf{w}^k(\mathbf{s}, \mathbf{a}) = 0$, in which case this transition is discarded and not used for learning anymore as the backup in Equation 7 preserves the Q-value (the second term) when $\mathbf{w}^k(\mathbf{s}, \mathbf{a}) = 0$, or (ii) the learning process reaches an equilibrium where $\mathbf{w}^k(\mathbf{s}, \mathbf{a}) = 1 \ \forall \ k$, meaning that this relabeled transition is used for learning.*

We have now provided a theoretical analysis of CUDS and a comparison between CUDS and UDS in Appendix F.2. Additionally, we provide several new experiments to build insight into why UDS and CUDS work in Appendix G.

### 4.3 PRACTICAL IMPLEMENTATIONS

We present pseudocode for UDS and CUDS in Algorithm 1. We train the Q-values with CQL to obtain conservative Q-values, and use the conservative Q-values to compute $\Delta^\pi(\mathbf{s}, \mathbf{a})$ defined in Eq. 2. For CUDS, in practice, instead of computing the hard threshold of $\Delta^\pi(\mathbf{s}, \mathbf{a}) \geq 0$ to determine data sharing, we follow Yu et al. (2021a) and transform the condition $\Delta^\pi(\mathbf{s}, \mathbf{a}) \geq 0$ into a soft weighting scheme, with weights given by $w_{\text{CUDS}}(\mathbf{s}, \mathbf{a}; j \to i) := \sigma \left( \frac{\Delta(\mathbf{s}, \mathbf{a}; j \to i)}{\tau} \right)$, where $\tau$ is a hyperparameter for the temperature of the sigmoid term in $w_{\text{CUDS}}$ that is automatically selected by the running average of $\Delta(\mathbf{s}, \mathbf{a}; j \to i)$. These weights are applied to both critic and actor training. For both UDS

and CUDS, we train a policy $\pi(\mathbf{a}|\mathbf{s}, i)$ where $\pi(\mathbf{a}|\mathbf{s}, i)$ could either be a single task-conditioned task with weight sharing or separate policies for each task without weight sharing. For more details of the practice implementations, see Appendix B.

## 5 EXPERIMENTS

In this section, we present our empirical evaluation, which aims to answer the following questions: (1) Can our simple approach outperform prior methods for utilizing unlabeled offline data on multi-task offline datasets? (2) Can the conservative data sharing strategy further improve the results achieved by our method? (3) Is our approach able to attain competitive result compared to the prior multi-task offline RL algorithm that have access to the true rewards? (4) How does CUDS compare to prior offline RL methods that directly learn representations from the multi-task offline dataset and run offline training on top of the representation?

**Comparisons.** We compare UDS and CUDS to a number of prior methods. We first evaluate: **No Sharing**, which performs applies standard offline RL algorithm to the multi-task setting without sharing data across tasks, **Reward Predictor**, which learns a classifier that directly predicts the reward using supervised learning, **VICE** (Fu et al., 2018c), an inverse RL method that learns a reward classifier from the labeled data and then annotates the unlabeled data with the learned classifier, and **RCE** (Eysenbach et al., 2021), a method similar to **VICE** except that **RCE** represents the Q-function as a classifier and learns the reward for unlabeled data implicitly. We adapt **VICE** and **RCE** to the multi-task offline RL setting by extracting transitions with reward labels equal to 1 and treating these datapoints as positives to learn the classifier for each task. We also train **VICE** and **RCE**, but adapt them to the offline setting using CQL, i.e. the same base offline RL method as in **UDS** and **CUDS**. Finally, to answer question (4), we conduct empirical evaluations on **ACL** (Yang & Nachum, 2021), which is a recent offline RL algorithm that performs representation learning on the offline dataset and trains the policy on top of the representation. For more details for experimental set-up and hyperparameter settings, please see Appendix B. We also include evaluations of our methods under different quality of the relabeled data in Appendix C, results of UDS and CUDS in dense-reward settings in Appendix D, comparisons to model-based offline RL approaches in Appendix E, and empirical analysis of the reasons that UDS and CUDS work in Appendix G.

### 5.1 MAIN EVALUATION

To answer questions (1), (2), and (3), we perform empirical evaluations on two state-based multi-task robotic manipulation and navigation datasets and one image-based multi-task manipulation dataset introduced in prior work (Yu et al., 2021a), which we will discuss below.

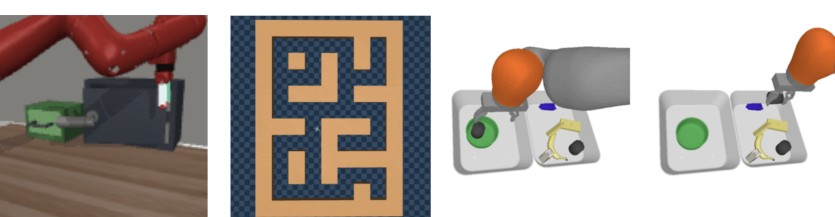

Figure 1: Environments (from left to right): Meta-World door and drawer open/close, AntMaze, and vision-based pick-place tasks.

**Tasks and Datasets.** Following the experimental setup in prior work (Yu et al., 2021a), we consider three domains shown in Fig. 1: (i) the Meta-World (Yu et al., 2020b) domain, which consists of four tasks of opening and closing doors and drawers; (ii) the Antmaze (Fu et al., 2020) domain, which consists of two sizes of mazes (medium and large) with 3 and 7 tasks respectively; and (iii) the multi-task visual manipulation domain, which consists of 10 tasks with different combinations of object-oriented grasping, with 7 objects (banana, bottle, sausage, milk box, food box, can and carrot), and placing the picked objects onto one of three fixtures (bowl, plate and divider plate). For all domains, we use binary rewards, where 1 denotes the successful completion of the task and 0 corresponds to failure. Note that for Meta-World, we use a fixed 200 timesteps for each episode and do not terminate the episode when receiving a reward of 1 at an intermediate timestep. In Antmaze, we terminate the episode upon seeing a reward of 1 with the maximum possible 1000 transitions per episode. We use the same datasets as prior work (Yu et al., 2021a). For Meta-World,

| Environment | Tasks | CUDS (ours) | UDS (ours) | VICE | RCE | No Sharing | Reward Predictor |
|---|---|---|---|---|---|---|---|
| Meta-World | door open | **61.3%**±7.9% | 51.9%±25.3% | 0.0%±0.0% | 0.0%±0.0% | 14.5%±12.7 | 0.0%±0.0% |
| | door close | 54.0%±42.5% | 12.3%±±47.1% | 66.7%%±47.1% | 0.0%±0.0% | 4.0%±6.1% | **99.3%**±0.9% |
| | drawer open | **73.5%**±9.6% | 61.8%±16.3% | 0.0%±0.0% | 0.0%±0.0% | 16.0%±17.5% | 13.3%±18.9% |
| | drawer close | 99.3%±0.7% | **99.6%**±0.7% | 19.3%±27.3% | 2.7%±1.7% | 99.0%±0.7% | 50.3%±35.8% |
| | average | **71.2%** ± 11.3% | 56.4%±12.8% | 21.5%±0.5% | 0.7%±0.4% | 33.4%±8.3% | 41.0%±11.9% |
| AntMaze | medium maze (3 tasks) | **31.5%**±3.0% | 26.5%±9.1% | 2.9%±1.0% | 0.0%±0.0% | 21.6%±7.1% | 3.8%±3.8% |
| | large maze (7 tasks) | **18.4%**±6.1% | 14.2%±3.9% | 2.5%±1.1% | 0.0%±0.0% | 13.3% ± 8.6% | 5.9%±4.1% |

Table 1: Results for multi-task robotic manipulation (Meta-World) and navigation environments (AntMaze) with low-dimensional state inputs. Numbers are averaged across 6 seeds, ± the 95%-confidence interval. We take the results of **No Sharing** directly from Yu et al. (2021a). We include per-task performance for Meta-World domains and the overall performance averaged across tasks (highlighted in gray) for all three domains. We bold the highest score across all methods. Both **CUDS** and **UDS** outperforms prior vanilla multi-task offline RL approach (**No Sharing**) and reward learning methods (**Reward Predictor**, **VICE** and **RCE**)

| Task Name | CUDS (ours) | UDS | No Sharing | CDS (oracle) | Sharing All (oracle) |
|---|---|---|---|---|---|
| lift-banana | **55.9%**±11.7% | 48.6%±5.1% | 20.0%±6.0% | **53.1%**±3.2% | 41.8%±4.2% |
| lift-bottle | **72.9%**±12.8% | 58.1%±3.6% | 49.7%±8.7% | **74.0%**±6.3% | 60.1%±10.2% |
| lift-sausage | **74.3%**±8.3% | 66.8% ± 2.7% | 60.9%±6.6% | **71.8%**±3.9% | 70.0%±7.0% |
| lift-milk | 73.5%±6.7% | **74.5%**±2.5% | 68.4%±6.1% | **83.4%**±5.2% | 72.5%±5.3% |
| lift-food | **66.3%**±8.3% | 53.8%±8.8% | 39.1%±7.0% | **61.4%**±9.5% | 58.5%±7.0% |
| lift-can | **64.9%**±7.1% | 61.0%±6.8% | 49.1%±9.8% | **65.5%**±6.9% | 57.7%±7.2% |
| lift-carrot | **84.1%**±3.6% | 73.4%±5.8% | 69.4%±7.6% | **83.8%**±3.5% | 75.2%±7.6% |
| place-bowl | **83.4%**±3.6% | 77.6%±1.6% | 80.3%±8.6% | **81.0%**±8.1% | 70.8%±7.8% |
| place-plate | **86.2%**±1.8% | 78.7%±2.2% | 86.1%±7.7% | **85.8%**±6.6% | 78.7%±7.6% |
| place-divider-plate | **89.0%**±2.2% | 80.2%±2.2% | 85.0%±5.9% | **87.8%**±7.6% | 79.2%±6.3% |
| average | **75.0%**±3.3% | 67.3%±0.8% | 60.8%±7.5% | **74.8%** ±6.4% | 66.4%±7.2% |

Table 2: Results for multi-task imaged-based robotic manipulation domains in (Yu et al., 2021a). Numbers are averaged across 3 seeds, ± the 95% confidence interval. **UDS** outperforms **No Sharing** in 7 out of 10 tasks as well as the average task performance, while performing comparably to **Sharing All**. **CUDS** further improves the performance of **UDS** and outperforms **No Sharing** in all of the 10 tasks.

we use large datasets with wide coverage of the state space and 152K transitions for the door open and drawer close tasks and datasets with limited (2K transitions), but optimal demonstrations for the door close and drawer open tasks. For AntMaze, following Yu et al. (2021a), we modify the datasets introduced by Fu et al. (2020) by equally dividing the large dataset into different parts for different tasks, where each task corresponds to a different goal position. For image-based manipulation, we directly use the dataset collected by Yu et al. (2021a), which contains a total of 100K RL episodes with 25 transitions for each episode, where the success rate is 40% and 80% for the picking and placing tasks, respectively. Note that the success rate of placing is higher because the robot is already holding the object at the start of the placing tasks, making the placing easier to solve.

**Results of Question (1).** The main results are in Table 1. **UDS** achieves better performance than vanilla multi-task offline RL without data sharing and compared to reward learning methods, suggesting that our simple relabeling method is effective in both multi-task manipulation and navigation domains. Since the reward learning approaches obtain similar or worse results compared to no sharing, we only compare our methods to **No Sharing** and the oracle methods in the image-based experiments. As shown in Table 2. **UDS** outperforms **No Sharing** in 7 out of 10 tasks as well as the average task performance by a significant margin. Therefore, **UDS** is able to effectively leverage unlabeled data shared from other tasks and achieves potentially surprisingly strong results compared to more sophisticated methods that handle unlabeled offline data, answering question (1).

**Results of Question (2).** In both the state-based and vision-based experiments shown in Table 1 and Table 2, we find that **CUDS** further improves upon the performance of **UDS**, which empirically indicates that the less conservative policy learned from CUDS's selective filtering scheme is more performant in practice. Additionally, we measure the success rates of the relabeled data in Table 5 Appendix C, measured by the oracle multi-task reward function on the Meta-World and AntMaze domain. We see that the success rates of the relabeled data are above 0% by a significant margin in most of the tasks. This suggests that UDS and CUDS are not simply relabeling with the true reward, since the relabeled data does not entirely consist of failures but rather has a significant number of successful transitions.

**Results of Question (3).** We also show results for oracle methods that receive true reward labels: **Sharing All**, which shares all data with ground truth rewards, and **CDS**, which uses the CDS strategy (Yu et al., 2021a) with ground truth reward relabeling. We present the results in Table 3 for state-based experiments and the last two columns on the right in Table 2 for the vision-based multi-

| Environment | Tasks | CUDS (ours) | UDS (ours) | CDS (oracle) | Sharing All (oracle) |
|---|---|---|---|---|---|
| Meta-World | door open | **61.3%**±7.9% | 51.9%±25.3% | 58.4%±9.3% | 34.3%±17.9% |
| | door close | 54.0% ±42.5% | 12.3%±27.6% | **65.3%**±27.7% | 48.3%±27.3% |
| | drawer open | **73.5%**±9.6% | 61.8%±16.3% | 57.9%±16.2% | 55.1%±9.4% |
| | drawer close | 99.3%±0.7% | 99.6%±0.7% | 99.0%±0.7% | 98.8%±0.7% |
| | average | **71.2%** ± 11.3% | 56.4%±12.8% | 70.1%±8.1% | 59.4%±5.7% |
| AntMaze | medium maze (3 tasks) | 31.5%±3.0% | 26.5%±9.1% | **36.7%**±6.2% | 22.9%±3.6% |
| | large maze (7 tasks) | 18.4%±6.1% | 14.2%±3.9% | **22.8%** ± 4.5% | 16.7% ± 7.0% |

Table 3: Comparison between **UDS / CUDS** and the oracle data sharing strategies with access to the true reward functions for relabeling. We take the results **CDS** and **Sharing All** directly from Yu et al. (2021a). **CDS** (Yu et al., 2021a) and **Sharing All** (Kalashnikov et al., 2021). **UDS / CUDS** achieve competitive results compared to **CUDS** and **UDS**.

| Environment | Tasks | CUDS (ours) | UDS (ours) | ACL |
|---|---|---|---|---|
| Meta-World | door open | **61.3%**±7.9% | 51.9%±25.3% | 2.8%±2.0% |
| | door close | 54.0% ±42.5% | 12.3%±27.6% | 0.0%±0.0% |
| | drawer open | 73.5%±9.6% | 61.8%±16.3% | **83.2%**±14.2% |
| | drawer close | 99.3%±0.7% | 99.6%±0.7% | **100.0%**±0.0% |
| | average | **71.2%** ± 11.3% | 56.4%±12.8% | 46.4%±3.5% |

Table 4: Comparison between **UDS / CUDS** and the **ACL** (Yang & Nachum, 2021) that performs representation learning on the unlabeled data instead of data sharing. Both **UDS** and **CUDS** outperforms **ACL** by a significant margin in the average task result, suggesting that sharing the unlabeled data is crucial in improving the multi-task offline RL performance compared to only using the data for learning the representation.

task robotic manipulation problem. Both **CUDS** and **UDS** achieves competitive results compared to **CDS** and **Sharing All**, indicating that our simple relabeling scheme is able to remove the dependence of functional form of reward functions without much loss of performance due to lacking ground-truth reward access. This addresses question (3).

**Results of Question (4).** Finally, to answer question (4), on the Meta-World environment, we compare UDS and CUDS to ACL (Yang & Nachum, 2021). We use the version of ACL without inputting reward labels. ACL can be viewed as an alternative to our unlabeled sharing data scheme, which leverages unlabeled data for representation learning rather than sharing it directly. We show the comparison to ACL in Table 4. UDS and CUDS outperform ACL in the average task performance while ACL is only proficient on drawer-open and drawer-close, and it cannot solve door-open or door-close. This indicates that sharing the unlabeled data conservatively across all tasks is important in multi-task offline RL while pretraining representations on the whole multi-task offline dataset might have limited benefit. We note that UDS / CUDS are complementary to ACL and these approaches can be combined together to further improve performance, which we leave as future work.

## 6 CONCLUSION

In the paper, we present two new algorithms, UDS and CUDS, that handle the problem of how to share data across tasks without access to the functional form of the multi-task reward function in the multi-task offline RL setting. UDS lifts the strong assumption of having access to the reward of all tasks at each transition in previous works in multi-task data sharing via simply sharing data across all tasks and relabeling the reward of data from other tasks to the minimum reward in the MDP, which indicates failure of the task. CUDS further improves over UDS via applying a more sophisticated data sharing scheme (Yu et al., 2021a) that shares data only if the relabeled Q-values improve over the expected Q-values of the original task data. We justify that UDS obtains Q-values that are lower-bounded by the Q-values learned by data sharing with true reward labels and then discuss that under certain structures of offline datasets, CUDS can selectively apply conservative policy evaluation on only transitions with high Q-values, resulting in a less conservative algorithm. Empirically, we show that both CUDS and UDS significantly outperform vanilla multi-task offline RL without data sharing as well as more complex methods that learns the reward function either explicitly or implicitly on a range of robotic manipulation and navigation domains. CUDS also improves over UDS on all of the domains. Furthermore, CUDS and UDS achieve competitive results compared to data sharing methods with access to the oracle rewards. While our method removes the strong assumption on reward functions in data sharing for multi-task offline RL and enjoys both theoretical guarantees and good empirical results, it does have a few limitations. For example, UDS and CUDS are evaluated in MDPs with binary rewards. Exploring their effects in MDPs with continuous rewards will be an exciting future avenue.

## REPRODUCIBILITY STATEMENT

We provided the code, data, and instructions needed to reproduce the main experimental results of the experiments with low-dimensional inputs in the supplementary material. We include all assumptions and derivations of all claims in Section 4.2 and Appendix A. For datasets used in our experiments, we discuss the details of processing the datasets in Appendix B.

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

## A  Q-VALUES LEARNED VIA UDS

**Proposition A.1.** *For a policy $\pi$, let $\widehat{Q}^\pi_{UDS}(\cdot, \cdot, i)$ denote the fixed point of Eqn. 6. Then, $\widehat{Q}^\pi_{UDS}(\mathbf{s}, \mathbf{a})$ lower-bounds the Q-function $\widehat{Q}^\pi_{Sharing\ All}$ that would be obtained had we used the true rewards.*

*Proof.* Note that $\widehat{Q}^\pi_{\mathrm{UDS}}(\mathbf{s}, \mathbf{a})$ at iteration $k+1$ is defined in Eq. 6, which we restate it for convenience:

$$\widehat{Q}^{k+1}_{\mathrm{UDS}}(\mathbf{s}, \mathbf{a}, i) \leftarrow \widehat{r}(\mathbf{s}, \mathbf{a}, i) + \gamma \mathbb{E}_{\mathbf{s}' \sim \widehat{P}(\mathbf{s}'|\mathbf{s}, \mathbf{a}), \pi(\mathbf{a}'|\mathbf{s}', i)} \left[ \widehat{Q}^k_{\mathrm{UDS}}(\mathbf{s}', \mathbf{a}', i) \right] - \alpha \left( \frac{\pi(\mathbf{a}|\mathbf{s}, i)}{\hat{\pi}_\beta(\mathbf{a}|\mathbf{s}, i)} - 1 \right), \quad (8)$$

whereas $\widehat{Q}^\pi_{\mathrm{Sharing\ All}}(\mathbf{s}, \mathbf{a})$ at iteration $k+1$ is defined as

$$\widehat{Q}^{k+1}_{\mathrm{Sharing\ All}}(\mathbf{s}, \mathbf{a}, i) \leftarrow r(\mathbf{s}, \mathbf{a}, i) + \gamma \mathbb{E}_{\mathbf{s}' \sim \widehat{P}(\mathbf{s}'|\mathbf{s}, \mathbf{a}), \pi(\mathbf{a}'|\mathbf{s}', i)} \left[ \widehat{Q}^k_{\mathrm{Sharing\ All}}(\mathbf{s}', \mathbf{a}', i) \right] \quad (9)$$

$$- \alpha \left( \frac{\pi(\mathbf{a}|\mathbf{s}, i)}{\hat{\pi}_\beta(\mathbf{a}|\mathbf{s}, i)} - 1 \right). \quad (10)$$

Assume $\widehat{Q}^0_{\mathrm{Sharing\ All}} = \widehat{Q}^0_{\mathrm{UDS}}$, i.e. same Q-value initialization and $\widehat{Q}^k_{\mathrm{Sharing\ All}} \geq \widehat{Q}^k_{\mathrm{UDS}}$. Using such induction hypothesis and the fact that $\widehat{r}(\mathbf{s}, \mathbf{a}, i) \leq r(\mathbf{s}, \mathbf{a}, i)$ for all $\mathbf{s}, \mathbf{a}$, we can conclude that $\widehat{Q}^\pi_{\mathrm{UDS}}(\mathbf{s}, \mathbf{a}) \leq \widehat{Q}^\pi_{\mathrm{Sharing\ All}}(\mathbf{s}, \mathbf{a})$. Therefore, UDS learns the Q-value that lower-bounds the Q-values learned data sharing all tasks with the ground-truth rewards. $\qquad\square$

## B  DETAILS OF UDS AND CUDS

In this section, we include the details of training UDS and CUDS in Appendix B.1 as well as details on the environment and datasets used in our experiments in Appendix B.2. Finally, we discuss the compute information of UDS and CUDS in Appendix B.3. For additional details, please see our anonymous website: https://sites.google.com/view/uds-cuds/.

### B.1  DETAILS ON THE TRAINING PROCEDURE

Our practical implementation of UDS optimizes the following objectives for the Q-functions and the policy:

$$\hat{Q}^{k+1} \leftarrow \arg\min_{\hat{Q}} \mathbb{E}_{i \sim [N]} \left[ \beta \left( \mathbb{E}_{j \sim [N]} \left[ \mathbb{E}_{\mathbf{s} \sim \mathcal{D}_j, \mathbf{a} \sim \mu(\cdot|\mathbf{s}, i)} \left[ \hat{Q}(\mathbf{s}, \mathbf{a}, i) \right] \right.\right.\right.$$
$$\left.\left.\left. - \mathbb{E}_{\mathbf{s}, \mathbf{a} \sim \mathcal{D}_j} \left[ \hat{Q}(\mathbf{s}, \mathbf{a}, i) \right] \right] \right) \right.$$
$$\left. + \frac{1}{2} \mathbb{E}_{j \sim [N], (\mathbf{s}, \mathbf{a}, \mathbf{s}') \sim \mathcal{D}_j} \left[ \left( \hat{Q}(\mathbf{s}, \mathbf{a}, i) - \left( r(\mathbf{s}, \mathbf{a}, i) \mathbb{1}_{\{j=i\}} + \gamma Q(\mathbf{s}', \mathbf{a}') \right) \right)^2 \right] \right],$$

and     $$\pi \leftarrow \arg\max_{\pi'} \mathbb{E}_{i \sim [N]} \left[ \mathbb{E}_{j \sim [N], \mathbf{s} \sim \mathcal{D}_j, \mathbf{a} \sim \pi'(\cdot|\mathbf{s}, i)} \left[ \hat{Q}^\pi(\mathbf{s}, \mathbf{a}, i) \right] \right],$$

Similarly, CUDS optimizes the following objectives for training the critic and the policy with a soft weight:

$$\hat{Q}^{k+1} \leftarrow \arg\min_{\hat{Q}} \mathbb{E}_{i \sim [N]} \left[ \beta \left( \mathbb{E}_{j \sim [N]} \left[ \mathbb{E}_{\mathbf{s} \sim \mathcal{D}_j, \mathbf{a} \sim \mu(\cdot|\mathbf{s}, i)} \left[ w_{\mathrm{CUDS}}(\mathbf{s}, \mathbf{a}; j \to i) \hat{Q}(\mathbf{s}, \mathbf{a}, i) \right] \right.\right.\right.$$
$$\left.\left.\left. - \mathbb{E}_{\mathbf{s}, \mathbf{a} \sim \mathcal{D}_j} \left[ w_{\mathrm{CUDS}}(\mathbf{s}, \mathbf{a}; j \to i) \hat{Q}(\mathbf{s}, \mathbf{a}, i) \right] \right] \right) \right.$$
$$\left. + \frac{1}{2} \mathbb{E}_{j \sim [N], (\mathbf{s}, \mathbf{a}, \mathbf{s}') \sim \mathcal{D}_j} \left[ w_{\mathrm{CUDS}}(\mathbf{s}, \mathbf{a}; j \to i) \left( \hat{Q}(\mathbf{s}, \mathbf{a}, i) - \left( r(\mathbf{s}, \mathbf{a}, i) \mathbb{1}_{\{j=i\}} + \gamma Q(\mathbf{s}', \mathbf{a}') \right) \right)^2 \right] \right],$$

and     $$\pi \leftarrow \arg\max_{\pi'} \mathbb{E}_{i \sim [N]} \left[ \mathbb{E}_{j \sim [N], \mathbf{s} \sim \mathcal{D}_j, \mathbf{a} \sim \pi'(\cdot|\mathbf{s}, i)} \left[ w_{\mathrm{CDS}}(\mathbf{s}, \mathbf{a}; j \to i) \hat{Q}^\pi(\mathbf{s}, \mathbf{a}, i) \right] \right],$$

where $\beta$ is the coefficient of the CQL penalty on distribution shift, $\mu$ is an action sampling distribution that covers the action bound as in CQL. We follow all the CQL hyperparameters used in Yu et al. (2021a).

To compute the weight $w_{\text{CUDS}}(\mathbf{s}, \mathbf{a}; j \to i) := \sigma\left(\frac{\Delta(\mathbf{s}, \mathbf{a}; j \to i)}{\tau}\right)$, we pick $\tau$, i.e. the temperature term, using the exponential running average of $\Delta(\mathbf{s}, \mathbf{a}; j \to i)$ with decay $0.995$ for each task following Yu et al. (2021a). Following Yu et al. (2021a) again, we clip the automatically chosen $\tau$ with a minimum and maximum threshold, which we directly use the values from Yu et al. (2021a). We use $[1, 50]$ and $[10, \infty]$ as the minimum and maximum threshold for the multi-task Meta-World and AntMaze domains respectively whereas the vision-based robotic manipulation domain does not require such clipping.

Following the training protocol in Yu et al. (2021a), for experiments with low-dimensional inputs, we use a stratified batch with $128$ transitions for each task to train the Q-functions and the policy. We also balance the numbers of transitions sampled from the original task and the number of transitions drawn from other task data. Specifically, for each task $i$, we sample $64$ transitions from $\mathcal{D}_i$ and the remaining $64$ transitions from $\cup_{j \neq i} \mathcal{D}_{j \to i}$. In CUDS, for each task $i \in [N]$, we only apply $w_{\text{CUDS}}$ to data shared from other tasks on multi-task Meta-World environments and multi-task vision-based robotic manipulation tasks while we also apply the relabeling weight to transitions sampled from the original task dataset $\mathcal{D}_i$ with 50% probability in the multi-task AntMaze domain.

Regarding the choices of the architectures, for state-based domains, we use 3-layer feedforward neural networks with $256$ hidden units for both the Q-networks and the policy. We condition the policy on a one-hot task ID, which is appended to the input state. In domains with high-dimensional image inputs, we adopt the multi-headed convolutional neural networks used in Kalashnikov et al. (2021); Yu et al. (2021a). We use images with dimension $472 \times 472 \times 3$, extra state features $(g_{\text{robot\_status}}, g_{\text{height}})$ and the one-hot task vector as the observations similar Kalashnikov et al. (2021); Yu et al. (2021a). Following the set-up in Kalashnikov et al. (2018; 2021); Yu et al. (2021a), we use Cartesian space control of the end-effector of the robot in 4D space (3D position and azimuth angle) along with two binary actions to open/close the gripper and terminate the episode respectively to represent the actions. For more details, see Kalashnikov et al. (2018; 2021).

## B.2 DETAILS ON THE ENVIRONMENT AND THE DATASETS

In this subsection, we include the discussion of the details the environment and datasets used for evaluating UDS and CUDS. Note that all of our environment and offline datasets are from prior work (Yu et al., 2021a). We will nonetheless discuss the details to make our work self-contained. We acknowledge that all datasets with low-dimensional inputs are under the MIT License.

**Multi-task Meta-World domains.** We use the `door open`, `door close`, `drawer open` and `drawer close` environments introduced in Yu et al. (2021a) from the public Meta-World (Yu et al., 2020c) repo[1]. In this multi-task Meta-World environment, a door and a drawer are put on the same scene, which ensures that all four tasks share the same state space. The environment uses binary rewards for each task, which are adapted from the success condition defined in the Meta-World public repo. In this case, the robot gets a reward of 1 if it solves the target task and 0 otherwise.

We direct use the offline datasets constructed in Yu et al. (2021a), which are generated by training online SAC policies for each task with the dense reward defined in the Meta-World repo for 500 epochs. The medium-replay datasets use the whole replay buffer of the online SAC agent until 150 epochs while the expert datasets are collected by the final online SAC policy.

**Multi-task AntMaze domains.** Following Yu et al. (2021a), we use the `antmaze-medium-play` and `antmaze-large-play` datasets from D4RL (Fu et al., 2020) and partitioning the datasets into multi-task datasets in an undirected way defined in Yu et al. (2021a). Specifically, the dataset is randomly splitted into chunks with equal size, and then each chunk is assigned to a randomly chosen task. Therefore, under such a task construction scheme, the task data for each task is of low success rate for the particular task it is assigned to and it is imperative for the multi-task offline RL algorithm to leverage effective data sharing strategy to achieve good performance. In AntMaze, we also use a binary reward, which provides the agent a reward of +1 when the ant reaches a position within a 0.5 radius of the task goal, which is also the reward used default by Fu et al. (2020). The terminal of an episode is set to be true when a reward of +1 is observed.

---

[1]The Meta-World environment can be found at the open-sourced repo `https://github.com/rlworkgroup/metaworld`

| Environment | Tasks | Oracle Success Rate of the Shared data |
|---|---|---|
| Meta-World | drawer open | 47.4% |
| | door close | 99.2% |
| | drawer open | 0.1% |
| | drawer close | 91.6%% |
| | **average** | 59.5% |
| AntMaze | medium maze (3 tasks) average | 4.3% |
| | large maze (7 tasks) average | 1.6% |

Table 5: Success rate of the data shared from other tasks to the target task determined by the ground-truth multi-task reward function.

**Multi-task image-based robotic picking and placing domains.** Following Kalashnikov et al. (2021); Yu et al. (2021a), we use sparse rewards for each task. That is, reward 1 is assigned to episodes that meet the success conditions and 0 otherwise. The success conditions are defined in (Kalashnikov et al., 2021). We directly use the dataset used in Yu et al. (2021a). Such a dataset is collected by first training a policy for each individual task using QT-Opt (Kalashnikov et al., 2018) until the success rate reaches 40% and 80% for picking tasks and placing tasks respectively and then combine the replay buffers of all tasks as the multi-task offline dataset. The dataset consists of a total number of 100K episodes with 25 transitions for each episode.

### B.3 COMPUTATION COMPLEXITY

We train UDS and CUDS on a single NVIDIA GeForce RTX 2080 Ti for one day on the state-based domains. For the vision-based robotic picking and placing experiments, it takes 3 days to train it on 16 TPUs.

## C ADDITIONAL DETAILS ON THE QUALITY OF DATA SHARED FROM OTHER TASKS

We present the success rate of the data shared from other tasks to the target task computed by the oracle multi-task reward function in both the multi-task Meta-World and AntMaze domains in Table 5. Note that the success rate of `drawer close` and `door close` are particularly high since for other tasks, the drawer / door is initialized to be closed and therefore the success rate of other task data for these two tasks are almost 100% as defined by the success condition in the public Meta-World repo. Apart from these two particularly high success rates, the success rates of the shared data are consistently above 0% across all tasks in both domains. This fact suggests that UDS and CUDS are *not* relabeling with the ground truth reward where the relabeled data are actually all failures but rather performs the conservative bellman backups on relabeled data that is shown to be effective empirically.

To better understand the performance of UDS under different relabeled data quality, we evaluate the UDS under different success rates of the data relabeled from other tasks in the multi-task Meta-World domain. Specifically, we filter out data shared from other tasks to ensure that the success rates of the relabeled data are 5%, 50% and 90% respectively. We compare the results of UDS on such data compositions to the performance of UDS in Table 1 where the success rate of relabeled data is 59.6% as shown in Table 5. The full results are in Table 6. UDS on relabeled data with 50% and 90% success rates achieves similar results compared to original UDS whereas UDS on relabel data with 5% success rate is significantly worse. Hence, UDS can obtain good results in settings where the relabeled data is of high quality despite incurring high reward bias, but is not helpful in settings where the shared data is of low quality and does not offer much information about solving the target task.

## D EMPIRICAL RESULTS OF UDS AND CUDS IN MORE GENERAL DENSE REWARD SETTINGS

In this section, we evaluate UDS and CUDS in the dense reward setting in order to test if UDS and CUDS work in more general reward settings and are not limited to binary rewards. We pick the multi-task walker environment as used in prior work (Yu et al., 2021a), which consists of three tasks, `run forward`, `run backward` and `jump`. The reward functions of the three tasks are $r(s, a) =$

| Environment | Tasks | UDS | UDS-5% relabel success | UDS-50% relabel success | UDS-90% relabel success |
|---|---|---|---|---|---|
| Meta-World | drawer open | 51.9%±25.3 | 0.0%±0.0% | 57.3%±18.9% | 73.3%±8.6% |
| | door close | 12.3%±27.6% | 0.0%±0.0% | 0.0%±0.0% | 0.0%±0.0% |
| | drawer open | 61.8%±16.3% | 19.4%±27.3% | 61.0%±12.7% | 56.3%±20.3% |
| | drawer close | 99.6%±0.7% | 66.0%±46.7% | 99.7%±0.5% | 100.0%±0.0% |
| | **average** | 56.4%±12.8% | 21.4% ±16.1% | 54.3% ±2.0% | 57.4%±3.3% |

Table 6: Performance of UDS under different actual success rates of the relabeled data.

| Environment | Tasks / Dataset type | CUDS (ours) | UDS (ours) | No Sharing | CDS (oracle) | Sharing All (oracle) |
|---|---|---|---|---|---|---|
| walker2d | run forward / medium-replay | 880.1±108.8 | 665.0±84.9 | 590.1±48.6 | **1057.9**±121.6 | 701.4±47.0 |
| | run backward / medium | 717.8±78.3 | 689.3±16.3 | 614.7±87.3 | 564.8±47.7 | **756.7**±76.7 |
| | jump / expert | 1487.7±177.6 | 1036.0±247.1 | **1575.2**±70.9 | 1418.2±138.4 | 885.1±152.9 |
| | **average** | **1028.6**±76.8 | 796.7±106.3 | 926.6±37.7 | 1013.6±71.5 | 781.0±100.8 |

Table 7: Results for multi-task walker experiment with dense rewards. CUDS and UDS are able to outperform No Sharing while attaining competitive results compared to CDS and Sharing All with oracle rewards. This suggests that CUDS and UDS are able to solve more general problems where rewards are not binary.

| Environment | Tasks | CUDS (ours) | UDS (ours | COMBO (Yu et al., 2021b) |
|---|---|---|---|---|
| Meta-World | door open | **61.3%**±7.9% | 51.9%±25.3 | 0.0%±0.0% |
| | door close | **54.0%** ±42.5% | 12.3%±27.6% | 1.1%±1.6% |
| | drawer open | **73.5%**±9.6% | 61.8%±16.3% | 15.7%±15.2% |
| | drawer close | 99.3%±0.7% | **99.6%**±0.7% | 85.7%±13.3% |
| | **average** | **71.2%** ± 11.3% | 56.4%±12.8% | 25.6%±6.2% |

Table 8: On the multi-task Meta-World domain, we compare CUDS and UDS to the model-based offline RL method COMBO (Yu et al., 2021b) that trains a dynamics model on all of the data and performs model-based offline training using the learned model. CUDS and UDS are able to outperform COMBO by a large margin.

$v_x - 0.001 * \|a\|_2^2$, $r(s, a) = -v_x - 0.001 * \|a\|_2^2$ and $r(s, a) = -\|v_x\| - 0.001 * \|a\|_2^2 + 10 * (z - \text{init z})$ respectively where $v_x$ denotes the velocity along the x-axis and $z$ denotes the z-position of the half-cheetah and init z denotes the initial z-position. In UDS and CUDS, we relabel the rewards routed from other tasks with the minimum reward value in the offline dataset of the target task. As shown in Table 7, CUDS and UDS outperform No Sharing by a large margin while also performing comparably to CDS and Sharing All. Therefore, CUDS and UDS are not limited to settings with binary rewards but are able to be applied to more general cases, in particular, environments with dense rewards.

# E  COMPARISONS OF CUDS AND UDS TO MULTI-TASK MODEL-BASED OFFLINE RL APPROACHES

In this section, we compare CUDS and UDS to a recent, state-of-the-art model-based offline RL method COMBO (Yu et al., 2021b) in the Meta-World domain. We adapt COMBO to the multi-task offline setting by learning the dynamics model on data of all tasks combined and and performing vanilla multi-task offline training without data sharing using the model learned with all of the data. As shown in Table 8, CUDS and UDS indeed outperform COMBO in the average task success rate. The intuition behind this is that COMBO is unable to learn an accurate dynamics model for tasks with limited data as in our Meta-World setting.

# F   THEORETICAL ANALYSIS OF UDS AND CUDS

In this section, we will theoretically analyze UDS and CUDS to understand when these approaches can perform well. We will first discuss our notation, then present our theoretical results, then discuss the intuitive explanations of these results, and finally, provide proofs of the theoretical results.

## F.1   NOTATION AND ASSUMPTIONS

Let $\pi_\beta(\mathbf{a}|\mathbf{s})$ denote the behavior policy for task $i$ (note that index $i$ was dropped from $\pi_\beta(\mathbf{a}|\mathbf{s}; i)$ for brevity). The dataset, $\mathcal{D}_i$ is generated from the marginal state-action distribution of $\pi_\beta$, i.e., $\mathcal{D} \sim d^{\pi_\beta}(\mathbf{s})\pi_\beta(\mathbf{a}|\mathbf{s})$. We define $d_\mathcal{D}^\pi$ as the state marginal distribution introduced by the dataset $\mathcal{D}$ under $\pi$. For our analysis, we will abstract offline RL algorithms into a generic constrained policy optimization problem (Kumar et al., 2020):

$$\pi^*(\mathbf{a}|\mathbf{s}) := \arg\max_\pi \ J_\mathcal{D}(\pi) - \frac{\alpha}{1-\gamma}D(\pi, \pi_\beta). \tag{11}$$

$J_\mathcal{D}(\pi)$ denotes the average return of policy $\pi$ in the empirical MDP induced by the transitions in the dataset, and $D(\pi, \pi_\beta)$ denotes a divergence measure (e.g., KL-divergence (Jaques et al., 2019; Wu et al., 2019), MMD distance (Kumar et al., 2019) or $D_{\text{CQL}}$ (Kumar et al., 2020)) between the learned policy $\pi$ and the behavior policy $\pi_\beta$. Let $D_{\text{CQL}}(p, q)$ denote the following distance between two distributions $p(\mathbf{x})$ and $q(\mathbf{x})$ with equal support $\mathcal{X}$:

$$D_{\text{CQL}}(p, q) := \sum_{\mathbf{x} \in \mathcal{X}} p(\mathbf{x}) \left( \frac{p(\mathbf{x})}{q(\mathbf{x})} - 1 \right).$$

Unless otherwise mentioned, we will drop the subscript "CQL" from $D_{\text{CQL}}$ and use $D$ and $D_{\text{CQL}}$ interchangeably. Prior works (Kumar et al., 2020; Yu et al., 2021a) have shown that the optimal policy $\pi_i^*$ that optimizes Equation 11 attains a high probability safe-policy improvement guarantee, i.e., $J(\pi_i^*) \geq J(\pi_\beta) - \zeta_i$, where $\zeta_i$ is:

$$\zeta_i = \mathcal{O}\left( \frac{1}{(1-\gamma)^2} \right) \mathbb{E}_{\mathbf{s} \sim d_{\mathcal{D}_i}^{\pi_i^*}} \left[ \sqrt{ \frac{D_{\text{CQL}}(\pi_i^*, \pi_\beta)(\mathbf{s}) + 1}{|\mathcal{D}_i(\mathbf{s})|} } \right] - \frac{\alpha}{1-\gamma}D(\pi_i^*, \pi_\beta). \tag{12}$$

The first term in Equation 12 corresponds to the decrease in performance due to sampling error and this term is high when the single-task optimal policy $\pi_i^*$ visits rarely observed states in the dataset $\mathcal{D}_i$ and/or when the divergence from the behavior policy $\pi_\beta$ is higher under the states visited by the single-task policy $\mathbf{s} \sim d_{\mathcal{D}_i}^{\pi_i^*}$. We will show that UDS and CUDS enjoy safe policy improvement. In our analysis, we assume $r(\mathbf{s}, \mathbf{a}) \in [0, 1]$. Finally, as discussed in Section 3, let $\mathcal{D}_i^{\text{eff}}$ denote the relabeled dataset for task $i$, which includes both $\mathcal{D}_i$ and the transitions from other tasks relabeled with a 0 reward.

**Assumptions.** To prove our theoretical results, following prior work (Kumar et al., 2020; Yu et al., 2021a) we assume that the empirical rewards and dynamics concentrate towards their mean.

**Assumption F.1.** $\forall \ \mathbf{s}, \mathbf{a}$, *the following relationships hold with high probability,* $\geq 1 - \delta$

$$|\widehat{r}(\mathbf{s}, \mathbf{a}) - r(\mathbf{s}, \mathbf{a})| \leq \frac{C_{r,\delta}}{\sqrt{|\mathcal{D}(\mathbf{s}, \mathbf{a})|}}, \quad ||\widehat{P}(\mathbf{s}'|\mathbf{s}, \mathbf{a}) - P(\mathbf{s}'|\mathbf{s}, \mathbf{a})||_1 \leq \frac{C_{P,\delta}}{\sqrt{|\mathcal{D}(\mathbf{s}, \mathbf{a})|}}.$$

Similar to prior work (Kumar et al., 2020; Yu et al., 2021a), we also make a coverage assumption, i.e., we assume that each state-action pair is observed in the dataset $\mathcal{D}_i$, but the rewards and transition dynamics are stochastic, so, the occurrence of each state-action pair does not trivially imply good performance. To relax this assumption, we can extend our analysis to function approximation (e.g., linear function approximation (Duan et al., 2020)), where such a coverage assumption is only required on all directions of the feature space, and not all state-action pairs. This would not significantly change the analysis, and hence we opt for the simple but illustrative analysis in a tabular setting here.

## F.2   THEORETICAL RESULTS

We first provide a performance guarantee for UDS which is then used to show that under certain conditions on the sizes of the labeled $\mathcal{D}_i$ and the effective dataset, $\mathcal{D}_i^{\text{eff}}$, UDS attains a better policy improvement guarantee than naïve no sharing. We first briefly discuss a novel component of our proof technique, then present the theoretical results, and then interpret it.

### F.2.1 OUR PROOF TECHNIQUE

While there are several techniques to provide guarantees for offline RL algorithms, we will build on the line of safe-policy improvement bounds, previously used in Kumar et al. (2020); Yu et al. (2021a). However, naïvely applying these guarantees to our UDS setting will give rise to very weak bounds, since a number of these guarantees utilize a bound on the value difference of the policy in the empirical MDP and the actual MDP (term (i)) as shown below:

$$J(\pi) - J(\pi_\beta) := \underbrace{J(\pi) - \widehat{J}(\pi)}_{(i)} + \underbrace{\widehat{J}(\pi) - \widehat{J}(\pi_\beta)}_{(ii)} + \underbrace{\widehat{J}(\pi_\beta) - \widehat{J}(\pi_\beta)}_{(iii)}.$$

Typically, term (i) depends on the sampling error on states that are visited by the learned policy $\pi$, and decays to 0 with infinite samples, but UDS can learn quite pessimistic Q-values due to the reward labeling procedure. However, this may not affect the policy performance since the relative ordering of actions might still be the same. This is not accounted for in any prior analysis we are aware of.

Therefore, we introduce a novel analysis tool that, rather than decomposing $J(\pi) - J(\pi_\beta)$ naïvely using the return in the empirical MDP, decomposes it using the return of the policy $\pi$ in the best empirical MDP that still produces the policy $\pi$ as its optimal policy. One simple way to obtain this best empirical MDP is via affine transformations on the reward function that preserve optimality. So, for strengthening our bound, we shall compute the bound similar to the above equation for different affine transformations of the reward and pick the one that gives the tightest bound.

Formally, let $g(\cdot)$ be an affine function: $g(x) = u \cdot x + v$ for some $u > 0, u \in \mathbb{R}$ and $v \in \mathbb{R}$. Then our decomposition looks like:

$$J(\pi) - J(\pi_\beta) := \underbrace{J(\pi) - g\left(\widehat{J}(\pi)\right)}_{(i)} + \underbrace{g\left(\widehat{J}(\pi)\right) - g\left(\widehat{J}(\pi_\beta)\right)}_{(ii)} + \underbrace{g\left(\widehat{J}(\pi_\beta)\right) - J(\pi_\beta)}_{(iii)},$$

Then, to obtain a strong lower bound on $J(\pi) - J(\pi_\beta)$, we can first bound each of the terms for a given choice of $g = (u, v)$, and then take the supremum over $u$ and $v$. This is reflected in the performance guarantee we present next.

### F.2.2 PERFORMANCE GUARANTEE FOR UDS

**Proposition F.1** (Policy improvement guarantee for UDS). *Let $\pi_{UDS}^*$ denote the policy learned by UDS for a given task $i$, and let $\pi_\beta^{\mathrm{eff}}(\mathbf{a}|\mathbf{s}, i)$ denote the behavior policy for the combined dataset for task $i$, $\mathcal{D}_i^{\mathrm{eff}}$. Then with high probability $\geq 1 - \delta$, $\pi_{UDS}^*$ is a $\zeta$-safe policy improvement over $\pi_\beta^{\mathrm{eff}}$, i.e., $J(\pi_{UDS}^*) \geq J(\pi_\beta^{\mathrm{eff}}) - \zeta$, where $\zeta$ is:*

$$\zeta = \min_{u>0, v} \ \zeta_{u,v}$$

$$\zeta_{u,v} = \underbrace{\frac{1}{1-\gamma}\left|\mathbb{E}_{\mathbf{s},\mathbf{a}\sim d_{\mathcal{D}_i^{\mathrm{eff}}}^{\pi_\beta}}\left[1 - f(\mathbf{s},\mathbf{a})\right] - v\right|}_{(a): \ reward \ bias, \ but \ modified \ for \ the \ best \ u} - \underbrace{\frac{\alpha u}{1-\gamma}D(\pi_{UDS}^*, \pi_\beta^{\mathrm{eff}})}_{(b): \ policy \ improvement}$$

$$+ \underbrace{\frac{2C_{P,\delta}\gamma}{(1-\gamma)^2}\left[\sqrt{\frac{D_{CQL}(\pi_{UDS}^*, \pi_\beta^{\mathrm{eff}})(\mathbf{s}) + 1}{|\mathcal{D}_i^{\mathrm{eff}}(\mathbf{s})|}}\right]}_{(c): \ dynamics \ sampling \ error} + \underbrace{\frac{2uC_{r,\delta}}{(1-\gamma)}\mathbb{E}_{\mathbf{s},\mathbf{a}\sim d_{\mathcal{D}_i}^{\pi}}\left[\frac{f(\mathbf{s},\mathbf{a})}{\sqrt{|\mathcal{D}_i(\mathbf{s},\mathbf{a})|}}\right]}_{(d): \ reward \ sampling \ error, \ but \ scaled \ down},$$

*where we use the notation $f(\mathbf{s},\mathbf{a}) := \frac{|\mathcal{D}_i(\mathbf{s},\mathbf{a})|}{|\mathcal{D}_i^{\mathrm{eff}}(\mathbf{s},\mathbf{a})|}$.*

A proof of Proposition F.1 is provided in Appendix F.3. To intuitively interpret the various terms that appear, we note that term (b) corresponds to the standard policy improvement that arises as a result of using an offline RL algorithm, term (c) corresponds to sampling error that arises as a result of performing offline RL on the dynamics induced by a finite dataset, but note that this term depends on the size of the effective dataset, $\mathcal{D}_i^{\mathrm{eff}}$ and not only the labeled dataset $\mathcal{D}_i$ for the task. Term (a) corresponds to the bias incurred as a result of labeling various transitions with a 0 reward in the data, and term (d) corresponds to the sampling error in the reward function, under the assumption of a stochastic reward function.

### F.2.3  HOW DOES UDS COMPARE TO NO SHARING?

In the setting when no data is shared across tasks, we attain the guarantee shown in Equation 12. Comparing Proposition F.1 to this guarantee, we note that under some scenarios, UDS yields a tighter bound compared to No Sharing. Two such scenarios are given by:

1.  **Long-horizon tasks:** Consider a scenario where tasks have a long horizon $H = \frac{1}{1-\gamma}$ and $|\mathcal{D}_i^{\text{eff}}(\mathbf{s})| = H^2|\mathcal{D}_i(\mathbf{s})|$. In this case, dynamics sampling error term (term (c)) consists of one less factor of $H$ when UDS is utilized, compared to when it is not. Since the dynamics sampling error grows quadratically in the horizon, whereas other terms grow linearly, a reduction in this term by increasing sample size (i.e., denominator) can lead to a stronger guarantee for UDS than No Sharing. This reasoning does not even consider term (d), which can be trivially upper-bounded by the corresponding term for No Sharing, even though UDS reduces this term as well.

2.  **The fraction $f(\mathbf{s}, \mathbf{a})$ is identical for all state-action pairs in the labeled $\mathcal{D}_i$, i.e., the unlabeled dataset consists of equal proportions state-action pairs as the labeled dataset.** Consider an extreme case when the unlabeled dataset consists of the trajectories in the labeled dataset such that $f(\mathbf{s}, \mathbf{a}) = c_0$ for all state-action tuples, just not annotated with rewards. In this case, reward bias takes on a constant value across all the transitions in the dataset, and by virtue of utilizing $u$ and $v$ in our bound in Proposition F.1, we note that the overall effect of this reward bias disappears, since $v$ can compensate for this bias.

### F.2.4  EXTENSION TO CUDS

Finally, we extend Proposition F.1 to a performance guarantee for CUDS by integrating the technique above with the analysis from Yu et al. (2021a). To analyze CUDS, we consider the abstract model of the conservative data sharing scheme developed by Yu et al. (2021a). This model suggests that CUDS approximates the following optimization in the empirical MDP generated by the relabeled dataset:

$$(\pi^*(\mathbf{a}|\mathbf{s}, i), \pi_\beta^*(\mathbf{a}|\mathbf{s}, i)) := \arg \max_{\pi, \pi_\beta \in \Pi_{\text{relabel}}} \widehat{J}_{\mathcal{D}^{\text{eff}}}(\pi) - \frac{\alpha}{1-\gamma} D(\pi, \pi_\beta). \tag{13}$$

Now, utilizing Proposition F.1 and Proposition 5.1 from Yu et al. (2021a), we obtain the following guarantee for CUDS:

**Corollary F.1.** *Let $\pi_{CUDS}^*(\mathbf{a}|\mathbf{s}, i)$ be the optimal policy found by CUDS (Equation 13) and let $\pi_\beta^*(\mathbf{a}|\mathbf{s}, i)$ denote the behavior policy that optimizes Equation 13 for task $i \in [N]$. Then, with high probability $\geq 1 - \delta$, $\pi_{CUDS}^*$ is a $\zeta$-safe policy improvement over $\pi_\beta^*$, i.e., $J(\pi_{CUDS}^*) \geq J(\pi_\beta^*) - \zeta_{CUDS}$, where $\zeta_{CUDS}$ is given by:*

$$\zeta_{CUDS} = \min_{u > 0, v} \zeta_{u,v}$$

$$\zeta_{u,v} = \frac{1}{1-\gamma} \underbrace{\left| \mathbb{E}_{\mathbf{s}, \mathbf{a} \sim d_{\mathcal{D}_i^{\text{eff}}}^{\pi_\beta}} [1 - f(\mathbf{s}, \mathbf{a})] - v \right|}_{\text{(a): reward bias}} - \underbrace{\frac{\alpha u}{1-\gamma} D(\pi_{CUDS}^*, \pi_\beta^*)}_{\text{(b): policy improvement}}$$

$$+ \underbrace{\frac{2C_{P,\delta}\gamma}{(1-\gamma)^2} \left[ \sqrt{\frac{D_{CQL}(\pi_{CUDS}^*, \pi_\beta^*)(\mathbf{s}) + 1}{|\mathcal{D}_i^{\text{eff}}(\mathbf{s})|}} \right]}_{\text{(c): dynamics sampling error}} + + \underbrace{\frac{2uC_{r,\delta}}{(1-\gamma)} \mathbb{E}_{\mathbf{s}, \mathbf{a} \sim d_{\mathcal{D}_i}^\pi} \left[ \frac{f(\mathbf{s}, \mathbf{a})}{\sqrt{|\mathcal{D}_i(\mathbf{s}, \mathbf{a})|}} \right]}_{\text{(d): reward sampling error, but scaled down}}.$$

*Proof.* The proof of this proposition follows directly from the proof of Proposition F.1 with the exception that this argument must be applied against the optimized behavior policy $\pi_\beta^*$.  □

**Comparing the bounds for CUDS and UDS.** We now interpret the bound in Corollary F.1 comparatively against the bound in Proposition F.1. First note that since the abstract model of CUDS (Equation 13) optimizes the behavior policy $\pi_\beta^*$, we first note from Equation 14 of Yu et al. (2021a) that for any other behavior policy $\pi'$,

$$D(\pi_{CUDS}^*, \pi_\beta^*) \leq D(\pi_{CUDS}^*, \pi'). \tag{14}$$

This means that the numerator of the sampling error term (term (c)) is smaller when CUDS is utilized as compared to when UDS is utilized. In addition, since CUDS relabels unlabeled data from Equation 13, this scheme also increases the dataset size, increasing the denominator of term (c). On the other hand, note that while UDS increases the denominator $|\mathcal{D}_i^{\text{eff}}|$, it may also increase the distributional shift $D(\pi^*, \pi_\beta^{\text{eff}})$ appearing in the numerator of the sampling error term. Our practical version of CUDS (Equation 2), which approximates Equation 13 by relabeling only the top $k$ percentile of the unlabeled data based on the objective in Equation 2, gives us a control over the effective dataset size after relabeling $|\mathcal{D}_i^{\text{eff}}|$, while still ensuring a reduced value of $D(\pi^*, \pi_\beta^*)$, and is thus expected to reduce $\zeta$ compared to UDS.

**Intuitively**, note that the bounds in Proposition F.1 and Corollary F.1, guarantee safe-policy improvement over different base policies $\pi_\beta^{\text{eff}}$ vs $\pi_\beta^*$. Intuitively, we would expect that $J(\pi_\beta^*) \geq J(\pi_\beta^{\text{eff}})$ in practice, especially for a large $\alpha$, since CUDS optimizes the behavior policy towards high return, compared to simply relabeling all unlabeled transitions. Therefore, CUDS not only reduces $\zeta$ compared to UDS, but also, in practice, is expected to improve over $\pi_\beta^*$, which performs better than $\pi_\beta^{\text{eff}}$. Thus, we would expect CUDS to be better in practice compared to UDS.

### F.3 PROOF OF PROPOSITION F.1

As mentioned in the beginning of Section F.2.1, to strengthen the conventional safe policy improvement bound, we instead utilize a different form of loss decomposition of the improvement of the learned policy relative to the behavior policy with the affine transformation $g$:

$$J(\pi) - J(\pi_\beta) := \underbrace{J(\pi) - g\left(\widehat{J}(\pi)\right)}_{(i)} + \underbrace{g\left(\widehat{J}(\pi)\right) - g\left(\widehat{J}(\pi_\beta)\right)}_{(ii)} + \underbrace{g\left(\widehat{J}(\pi_\beta)\right) - J(\pi_\beta)}_{(iii)}.$$

Now we will discuss how to bound each of the terms: terms (i) and (ii) correspond to the divergence between a transformed empirical policy return and the actual return. While usually, this difference depends on the sampling error and distributional shift, in our case, it additionally depends on the reward bias induced on the unlabeled data and the transformation $g$. We first discuss the terms that contribute to this reward bias.

**Bounding the reward bias.** Denote the effective reward of a particular transition $(\mathbf{s}, \mathbf{a}, r, \mathbf{s}') \in \mathcal{D}_i^{\text{eff}}$, as $\widehat{r}_i^{\text{eff}}$, which considers contributions from both the reward $\widehat{r}(\mathbf{s}, \mathbf{a})$ observed in dataset $\mathcal{D}_i$, and the contribution of 0 reward from the relabeled dataset:

$$\widehat{r}_i^{\text{eff}}(\mathbf{s}, \mathbf{a}) = \frac{|\mathcal{D}_i(\mathbf{s}, \mathbf{a})| \cdot \widehat{r}(\mathbf{s}, \mathbf{a}) + |\mathcal{D}_i^{\text{eff}}(\mathbf{s}, \mathbf{a}) \setminus \mathcal{D}_i(\mathbf{s}, \mathbf{a})| \cdot 0}{|\mathcal{D}_i^{\text{eff}}(\mathbf{s}, \mathbf{a})|} \quad (15)$$

Define $f(\mathbf{s}, \mathbf{a}) := \frac{|\mathcal{D}_i(\mathbf{s},\mathbf{a})|}{|\mathcal{D}_i^{\text{eff}}(\mathbf{s},\mathbf{a})|}$ for notation compactness. Equation 15 and the form of the reward transformation $g(x) = u \cdot x + v$ can then be used to derive the following difference against the true rewards:

$$u\widehat{r}_i^{\text{eff}}(\mathbf{s}, \mathbf{a}) + v - r(\mathbf{s}, \mathbf{a}) = uf(\mathbf{s}, \mathbf{a})\left(\widehat{r}(\mathbf{s}, \mathbf{a}) - r(\mathbf{s}, \mathbf{a})\right) + (1 - uf(\mathbf{s}, \mathbf{a})) \cdot (0 - r(\mathbf{s}, \mathbf{a})) + v \quad (16)$$

$$\leq uf(\mathbf{s}, \mathbf{a}) \cdot \frac{C_{r,\delta}}{\sqrt{|\mathcal{D}_i(\mathbf{s}, \mathbf{a})|}} - (1 - f(\mathbf{s}, \mathbf{a})u) \cdot r(\mathbf{s}, \mathbf{a}) + v \quad (17)$$

$$\leq uf(\mathbf{s}, \mathbf{a}) \cdot \frac{C_{r,\delta}}{\sqrt{|\mathcal{D}_i(\mathbf{s}, \mathbf{a})|}},$$

where the last step follows from the fact that the ground-truth reward $r(\mathbf{s}, \mathbf{a}) \in [0, 1]$ and the fact that $v$ will be chosen to minimize this upper bound. Now, we lower bound the reward bias as follows:

$$u\widehat{r}_i^{\text{eff}}(\mathbf{s}, \mathbf{a}) + v - r(\mathbf{s}, \mathbf{a}) = uf(\mathbf{s}, \mathbf{a}) \cdot (\widehat{r}(\mathbf{s}, \mathbf{a}) - r(\mathbf{s}, \mathbf{a})) + (1 - f(\mathbf{s}, \mathbf{a})u) \cdot (-r(\mathbf{s}, \mathbf{a})) + v \quad (18)$$

$$\geq -uf(\mathbf{s}, \mathbf{a}) \cdot \frac{C_{r,\delta}}{\sqrt{|\mathcal{D}_i(\mathbf{s}, \mathbf{a})|}} - (1 - f(\mathbf{s}, \mathbf{a})u) + v,$$

where the last step follows from the fact that $r(\mathbf{s}, \mathbf{a}) \leq 1$. To highlight the significance of this reward transformation, note that in the last step, if $\forall \mathbf{s}, \mathbf{a}, \; f(\mathbf{s}, \mathbf{a}) = c_0$, then the best reward transformation would choose $v = 1 - c_0$, and that completely eliminates the excess bias induced in the bound.

**Upper bounding** $g\left(\widehat{J}_i(\pi)\right) - J_i(\pi)$**.** Next, using the upper and lower bounds on the reward bias, we now derive an upper bound on the difference between the value of a policy computed under the empirical MDP and the actual MDP. To compute this difference, we follow the following steps

$$g\left(\widehat{J}_i(\pi)\right) - J_i(\pi) = \frac{1}{1-\gamma}\sum_{\mathbf{s},\mathbf{a}}\left(\widehat{d}^{\pi}_{\mathcal{D}^{\text{eff}}_i}(\mathbf{s})\pi(\mathbf{a}|\mathbf{s})g\left(\widehat{r}^{\text{eff}}_i(\mathbf{s},\mathbf{a})\right) - d^{\pi}_i(\mathbf{s})\pi(\mathbf{a}|\mathbf{s})r(\mathbf{s},\mathbf{a})\right) \tag{19}$$

$$\leq \frac{1}{1-\gamma}\underbrace{\sum_{\mathbf{s},\mathbf{a}}\widehat{d}^{\pi}_{\mathcal{D}^{\text{eff}}_i}(\mathbf{s})\pi(\mathbf{a}|\mathbf{s})\left(g\left(\widehat{r}^{\text{eff}}_i(\mathbf{s},\mathbf{a})\right) - r(\mathbf{s},\mathbf{a})\right)}_{:=\Delta_1} + \frac{1}{1-\gamma}\underbrace{\sum_{\mathbf{s},\mathbf{a}}\left(\widehat{d}^{\pi}_{\mathcal{D}^{\text{eff}}_i}(\mathbf{s}) - d^{\pi}(\mathbf{s})\right)\pi(\mathbf{a}|\mathbf{s})r(\mathbf{s},\mathbf{a})}_{:=\Delta_2}$$

Following Kumar et al. (2020) (Theorem 3.6), we can bound the second term $\Delta_2$ using:

$$|\Delta_2| \leq \frac{\gamma C_{P,\delta}}{1-\gamma}\mathbb{E}_{\mathbf{s}\sim\widehat{d}^{\pi}_{\mathcal{D}^{\text{eff}}_i}(\mathbf{s})}\left[\frac{\sqrt{|\mathcal{A}|}}{\sqrt{|\mathcal{D}^{\text{eff}}(\mathbf{s})|}}\sqrt{D(\pi,\widehat{\pi}^{\text{eff}}_\beta)(\mathbf{s}) + 1}\right]. \tag{20}$$

To upper bound $\Delta_1$, we utilize the reward upper bound from Equation 16:

$$\Delta_1 = \sum_{\mathbf{s}}\widehat{d}^{\pi}_{\mathcal{D}^{\text{eff}}_i}(\mathbf{s})\left(\sum_{\mathbf{a}}\pi(\mathbf{a}|\mathbf{s})\left(u\widehat{r}^{\text{eff}}_i(\mathbf{s},\mathbf{a}) + v - r(\mathbf{s},\mathbf{a})\right)\right) \tag{21}$$

$$\leq \underbrace{\sum_{\mathbf{s}}\widehat{d}^{\pi}_{\mathcal{D}^{\text{eff}}_i}(\mathbf{s})\sum_{\mathbf{a}}uf(\mathbf{s},\mathbf{a})\frac{C_{r,\delta}}{\sqrt{|\mathcal{D}_i(\mathbf{s})|}}\frac{\pi(\mathbf{a}|\mathbf{s})}{\sqrt{\widehat{\pi}_\beta(\mathbf{a}|\mathbf{s})}}}_{=\Delta'_1}. \tag{22}$$

Combining the results so far, we obtain, for any policy $\pi$:

$$J_i(\pi) \geq g\left(\widehat{J}_i(\pi)\right) - \frac{|\Delta_2|}{1-\gamma} - \frac{|\Delta'_1|}{1-\gamma}. \tag{23}$$

**Lower bounding** $g\left(\widehat{J}_i(\pi)\right) - J_i(\pi)$**.** To lower bound this quantity, we follow the step shown in Equation 19, and lower bound the term $\Delta_2$ by using the negative of the RHS of Equation 20, and lower bound $\Delta_1$ by upper bounding its absolute value as shown below:

$$|\Delta_1| = \left|\sum_{\mathbf{s}}\widehat{d}^{\pi}_{\mathcal{D}^{\text{eff}}_i}(\mathbf{s})\left(\sum_{\mathbf{a}}\pi(\mathbf{a}|\mathbf{s})\left(u\widehat{r}^{\text{eff}}_i(\mathbf{s},\mathbf{a}) + v - r(\mathbf{s},\mathbf{a})\right)\right)\right| \tag{24}$$

$$\leq \underbrace{\sum_{\mathbf{s}}\widehat{d}^{\pi}_{\mathcal{D}^{\text{eff}}_i}(\mathbf{s})\sum_{\mathbf{a}}uf(\mathbf{s},\mathbf{a})\frac{C_{r,\delta}}{\sqrt{|\mathcal{D}_i(\mathbf{s})|}}\frac{\pi(\mathbf{a}|\mathbf{s})}{\sqrt{\widehat{\pi}_\beta(\mathbf{a}|\mathbf{s})}}}_{=\Delta'_1} + \left|\sum_{\mathbf{s}}\widehat{d}^{\pi}_{\mathcal{D}^{\text{eff}}_i}(\mathbf{s})\sum_{\mathbf{a}}\pi(\mathbf{a}|\mathbf{s})\cdot(1 - f(\mathbf{s},\mathbf{a})u) - v\right|.$$

$$\tag{25}$$

This gives rise to the complete lower bound:

$$g\left(\widehat{J}_i(\pi)\right) \geq J_i(\pi) - \frac{|\Delta_2|}{1-\gamma} - \frac{1}{1-\gamma}\left|\sum_{\mathbf{s},\mathbf{a}}\widehat{d}^{\pi}_{\mathcal{D}^{\text{eff}}_i}(\mathbf{s})\pi(\mathbf{a}|\mathbf{s})(1 - f(\mathbf{s},\mathbf{a})u) - v\right| - \frac{\Delta'_1}{1-\gamma}. \tag{26}$$

**Policy improvement term (ii).** Finally, the missing piece that needs to be bounded is the policy improvement term (ii) in the decomposition of $g\left(J(\pi)\right) - g\left(J(\pi_\beta)\right)$. Utilizing the abstract form of offline RL (Equation 11, we note that term (ii) is lower bounded as:

$$\text{term (ii)} \geq \frac{\alpha u}{1-\gamma}D(\pi,\pi_\beta). \tag{27}$$

**Putting it all together.** To obtain the final expression of Proposition F.1, we put all the parts together, and include some simplifications to obtain the final expression. The bound we show is relative to the

effective behavior policy $\pi_\beta^{\text{eff}}$. Applying Equation 26 for term (i) on policy $\pi$, Equation 27 for term (ii), and Equation 23 for the behavior policy $\pi_\beta^{\text{eff}}$, we obtain the following:

$$J(\pi) - J(\pi_\beta^{\text{eff}}) = J(\pi) - g\left(\widehat{J}(\pi)\right) + g\left(\widehat{J}(\pi)\right) - g\left(\widehat{J}(\pi_\beta^{\text{eff}})\right) + g\left(\widehat{J}(\pi_\beta^{\text{eff}})\right) - J(\pi_\beta^{\text{eff}})$$

$$\geq -\frac{2\gamma C_{P,\delta}}{(1-\gamma)^2}\mathbb{E}_{\mathbf{s}\sim\widehat{d}_{\mathcal{D}_i^{\text{eff}}}^\pi(\mathbf{s})}\left[\frac{\sqrt{|\mathcal{A}|}}{\sqrt{|\mathcal{D}^{\text{eff}}(\mathbf{s})|}}\sqrt{D(\pi,\widehat{\pi}_\beta^{\text{eff}})(\mathbf{s})+1}\right] - \frac{2uC_{r,\delta}}{1-\gamma}\mathbb{E}_{\mathbf{s},\mathbf{a}\sim\widehat{d}_{\mathcal{D}_i^{\text{eff}}}^\pi}\left[\frac{f(\mathbf{s},\mathbf{a})}{\sqrt{|\mathcal{D}_i(\mathbf{s},\mathbf{a})|}}\right]$$

$$-\underbrace{\frac{1}{1-\gamma}\left|\mathbb{E}_{\mathbf{s},\mathbf{a}\sim d_{\mathcal{D}_i^{\text{eff}}}^{\pi_\beta}}[1-f(\mathbf{s},\mathbf{a})u]-v\right|}_{:=\Delta_3} + \frac{\alpha u}{1-\gamma}D(\pi,\pi_\beta^{\text{eff}}).$$

Note that in the second step above, we upper bound the quantities $\Delta_1'$ and $\Delta_2$ corresponding to $\pi_\beta^{\text{eff}}$ with twice the expression for policy $\pi$. This is because the effective behavior policy $\pi_\beta^{\text{eff}}$ consists of a mixture of the original behavior policy $\widehat{\pi}_\beta$ with the additional data, and thus the new effective dataset consists of the original dataset $\mathcal{D}_i$ as its part. Upper bounding it with twice the corresponding term for $\pi$ is a valid bound, though a bit looser, but this bound suffices for our interpretations.

Finally to finish the proof, we can take the supremum over the best choice of $(u, v)$. Thus, we obtain the desired bound in Proposition F.1.

## G   EMPIRICAL ANALYSIS OF THE REASON THAT CUDS AND UDS WORK

In this section, we perform an empirical study on the Meta-World domain to better understand the reason that UDS and CUDS work well. Our theoretical analysis suggests that UDS will help the most on domains with limited data or narrow coverage or low data quality. To test these conditions in practice, we perform empirical analysis on two domains as follows.

### G.1   META-WORLD DOMAINS

We first choose the `door open` task with three different combinations of dataset size and data quality of the task-specific data with reward labels:

- 2k transitions with the expert-level performance (i.e. **high-quality data with limited sample size and narrow coverage**)

- 2k transitions with medium-level performance (i.e. **medium-quality data with limited sample size and narrow coverage**)

- a medium-replay dataset with 152k transitions (i.e. **medium-quality data with sufficient sample size and broad coverage**).

We share the same data from the other three tasks, `door close`, `drawer open` and `drawer close` as in Table 1, which are . As shown in Table 9, both UDS and CUDS are able to outperform No Sharing in the three settings, suggesting that increasing the coverage of the offline data as suggested by our theory does lead to performance boost in wherever we have limited good-quality data (expert), limited medium-quality data (medium) and abundant medium-quality data (medium-replay). It's worth noting that UDS and CUDS significantly outperform No Sharing in the limited expert and medium data setting whereas in the medium-replay setting with broader coverage, CUDS outperforms No sharing but UDS fails to achieve non-zero success rate. Such results suggest that UDS and CUDS can yield greater benefit when the target task doesn't have sufficient data and the number of relabeled data is large. The fact that UDS is unable to learn on medium-replay datasets also suggests that data sharing without rewards is less useful in settings where the coverage of the labeled offline data is already quite broad.

### G.2   D4RL HOPPER DATA QUALITY + COVERAGE DIAGNOSTIC STUDY

To further understand the sensitivity of UDS to the data coverage and the data quality of both target task data (i.e. with reward labels) and relabeled data (i.e. without reward labels), we perform another empirical study using the `hopper` environment from the D4RL (Fu et al., 2020) benchmark. We consider the following 6 different combinations varying the quality and amount of the labeled and unlabeled datasets:

| Environment | Dataset type / size | CUDS (ours) | UDS (ours | No Sharing |
|---|---|---|---|---|
| Meta-World door open | expert / 2k transitions | **67.6%** | 58.8% | 31.3% |
| | medium / 2k transitions | 67.3% | **74.2%** | 27.6% |
| | medium-replay / 152k transitions | **30.0%** | 0.0% | 14.8% |

Table 9: We perform an empirical analysis on the Meta-World `door open` task where we use varying data quality and dataset size target task `door open`. We share the same dataset from the other three tasks in the multi-task Meta-World environment, `door close`, `drawer open` and `drawer close` to the target task. The numbers are averaged over three random seeds. CUDS and UDS are able to outperform No Sharing in most of the settings except that UDS fails to achieve non-zero success rate in the medium-replay dataset with a large number of transitions. Such results suggest that CUDS and UDS are robust to the data quality of the target task and work the best in settings where the target task has limited data.

1. 10k labeled data from `hopper-expert` + unlabeled 1M data `hopper-random` (i.e., **high-quality + narrow labeled data, low-quality + broad unlabeled data**)

2. 10k labeled data from `hopper-expert` + unlabeled 1M data from `hopper-medium` (i.e., **high-quality + narrow labeled data, medium-quality + narrow unlabeled data**)

3. 10k labeled data from `hopper-medium` + unlabeled 1M data from `hopper-random` (i.e., **medium-quality + narrow labeled data, low-quality + broad unlabeled data**)

4. 10k labeled data from `hopper-medium` + unlabeled 1M data from `hopper-expert` (i.e., **medium-quality + narrow labeled data, high-quality + narrow unlabeled data**)

5. 10k labeled data from `hopper-random` + unlabeled 1M data from `hopper-medium` (i.e., **low-quality + broad labeled data, medium-quality + narrow unlabeled data**)

6. 10k labeled data from `hopper-random` + unlabeled 1M data from `hopper-expert` (i.e., **low-quality + broad labeled data, high-quality + narrow unlabeled data**)

**Results.** In cases (1) and (2), adding the unlabeled random or medium data, should increase coverage, since the labeled data only consists of expert transitions. Moreover, the induced reward bias due to incorrect labeling of rewards on the medium unlabeled data should not hurt, since the 10k expert transitions retain their correct labels, and the medium/random data should only serve as negatives. Therefore, we expect the benefits of coverage to outweigh any reward bias, and as shown in Table 10, we find that UDS does help.

In cases (4), (5) and (6), when the relabeled data is better compared to the labeled data (i.e., expert or medium), we find that even if the rewards on these transitions are incorrect, behavior regularization properties induced by offline RL algorithms allow UDS to attain better performance than no sharing by utilizing the unlabeled data.

In case (3), we find that UDS hurts compared to No Sharing. This is because the target task data as well as unlabeled data are both low-medium quality and medium data already provides decent coverage (not as high as random data, but not as low as expert data). Therefore, in this case, we believe that the addition of unlabeled data neither provides trajectories of good quality that can help improve performance, nor does it significantly improve coverage, and only hurts by incurring reward bias. We therefore believe that UDS may not help in such cases where the coverage does not improve, and added data is not so high quality.

### G.3 SUMMARY OF EMPIRICAL ANALYSIS

Given our results in Table 9 and Table 10, we summarize the applicability of UDS under different scenarios in Table 11 below.

| Environment | Labeled dataset type / size | Unlabeled dataset type / size | UDS (ours) | No Sharing |
|---|---|---|---|---|
| D4RL hopper (Fu et al., 2020) | expert / 10k transitions | random / 1M transitions | **90.8** | 77.1 |
| | expert / 10k transitions | medium / 1M transitions | **87.6** | 77.1 |
| | medium / 10k transitions | random / 1M transitions | 9.8 | **28.7** |
| | medium / 10k transitions | expert / 1M transitions | **106.**1 | 28.7 |
| | random / 10k transitions | medium / 1M transitions | **51.9** | 9.6 |
| | random / 10k transitions | expert / 1M transitions | **97.0** | 9.6 |

Table 10: We perform an empirical analysis on the `hopper` environment from the D4RL (Fu et al., 2020) benchmark to test the sensitivity of UDS under the data quality and data coverage for both the labeled task data and unlabeled data. The numbers are averaged over three random seeds. UDS outperforms No Sharing in 5 out of 6 settings, suggesting that UDS is robust in different combinations of data quality and coverage of both labeled and unlabeled data. Note that UDS fails in the setting where the labeled data is of medium data quality and the unlabeled data is random, suggesting that sharing data in settings where the labeled data is limited and of low quality and the unlabeled data is also of poor quality is not useful.

| Scenarios | UDS | Intuition |
|---|---|---|
| **L**: limited + high-quality + narrow, **U**: abundant + low-quality + broad | ✓ | increase coverage |
| **L**: limited + high-quality + narrow, **U**: abundant + medium-quality + narrow | ✓ | more negatives |
| **L**: limited + medium-quality + narrow, **U**: abundant + low-quality + broad | ✗ | reward bias outweighs high coverage |
| **L**: abundant + medium-quality + broad, **U**: abundant + medium-quality + broad | ✗ | reward bias outweighs high coverage |
| **L**: limited + medium-quality + narrow, **U**: abundant + high-quality + narrow | ✓ | increase data quality |
| **L**: limited + low-quality + broad, **U**: abundant + medium-quality + narrow | ✓ | increase data quality |
| **L**: limited + low-quality + broad, **U**: abundant + high-quality + narrow | ✓ | increase data quality |

Table 11: Summary of scenarios where UDS is expected to work and where it is not expected to work. **L** denotes the characteristics of labeled data, **U** denotes characteristics of unlabeled data. Limited/Abundant refers to the relative amount of data available (note that these are not absolute numbers and hard to precisely quantify without access to the problem domain, but a highly skewed ratio of the amount of labeled and unlabeled data might help characterize it as limited/abundant). High-quality/medium-quality/low-quality refers to the actual performance of the behavior policy generating the datasets. Narrow/broad refers to the relative state coverage of the datasets that we study.

