# OpenReview forum: "Data Sharing without Rewards in Multi-Task Offline Reinforcement Learning"
_ICLR.cc/2022/Conference — ICLR 2022 Submitted_

### Official Review · Reviewer_SFTg · 2021-10-31

**Correctness:** 4
**Technical Novelty And Significance:** 2
**Empirical Novelty And Significance:** 2
**Recommendation:** 5
**Confidence:** 4

**Main Review:**

Advantages:
1. This work proposed two easily implemented approaches called CUDS and UDS to achieve data sharing in offline reinforcement learning settings, without the requirement of a reward function approximation.

Some of the questions under concern:
1. One of the most important claims in this work is that under some assumptions, the proposed data sharing methods (CUDS, UDS) will be beneficial even suppress the oracle. However, there is not a clear statement or discussion about the intuition when and why it can achieve this? Although there are some experiments such as meta-world, etc., it is still confusing whether these data-sharing methods can be utilized in other application scenarios.
2. There is an assumption in the targeted MDP is that the reward is binary. Is it necessary for CUDS/UDS to work?
3. In section 4.2, the author tried to provide the intuition of how data sharing influences Q-estimates. However, remark 4.2 seems to be unclear and can't lead to the later conclusion that " CUDS does not have to push down Q-values at every state-action pair". Can you give more explanation about it?
4. The writing of the paper needs to be polished a lot. I will provide extra comments later.

Some minor comments about writing:
1. A lot of descriptions and equations are too similar to the most related work [1]? It may need more revision. Such as the equ 2 and 3, even Algorithm 1 looks all similar.
2. What's the definition of '[N]' and it has been used as a distribution in Section 3?
3. Typo in Section 3 as 'D>'?
4. The definition in Equ (1) is not clear? Such as what is (s, a), what is $\pi$, and what the $\mathcal{D}_i^{eff}$ may need the input of $\pi$, since different policies may lead to different sharing datasets?
5. More omits of definitions after equation (3), such as what's the exact formulation of the notation $J$ and $D(\pi,\pi_\beta)$
6. The text "CQL" is also introduced without reference, although it may refer to conservative Q-learning.
7. Undefined $\hat{Q}_{UDS}^\pi(s,a)$ in Proposition 4.1
8. $w$ is a binary matrix as defined above of equation (7), not a vector. Moreover, what is the difference between $w(s,a)$ and $w(s,a,I)$?

**Summary Of The Paper:**

This work targets useful data sharing between multi-tasks in an offline reinforcement learning setting. Motivated by the same transition kernel $P$, for each task, data sharing from other tasks is considered to be beneficial for finding an optimal policy. Different from previous works, the author claims that under some assumptions, data sharing using a constant reward/no reward can be better than using a reward predictor and competitive with using ground-truth rewards relabelled. Towards this, the author proposes two methods called CUDS and UDS and conducts experiments to test its performance.

**Summary Of The Review:**

I recommend that this paper be below the acceptance threshold.

1. This paper release the strong assumption with a reward relabelling function in [1], which archives data sharing between multi-tasks without reward. It seems like an extension with a minor revision of CDS in [1].
2. The intuition of the great performance is also straightforward since more data yields a better estimation of the transition kernel $P$ in MDP, even without rewards.
3. Although in the examined experiments the proposed methods show competitive performances compared to CDS and other baselines, it is not convincing enough that without an estimation of the reward, CUDS/UDS can achieve great results in more general scenarios.


[1] Yu, Tianhe, et al. "Conservative data sharing for multi-task offline reinforcement learning." arXiv preprint arXiv:2109.08128 (2021).

---

> ### Author Response · Authors · 2021-11-15
> **Author Response(Part 1): Summary of Changes, Differences from prior works, intuitions why the methods work**
>
> Thank you for the review! We believe the feedback has helped improve the paper, and we have attempted to address all the concerns raised in the review.
>
> To begin with, we address the concern regarding **novelty** -- while CUDS and UDS are indeed simple modifications to the existing CDS approach, we would like to emphasize that CUDS and UDS remove the assumption of having access to the oracle reward functions of each task, which is problematic in practice due to the hard reward specification problem. In fact, our empirical results suggest that UDS and CUDS can outperform sophisticated reward-learning methods such as VICE and RCE indicating that UDS and CUDS are simple, yet effective. We believe that the empirical novelty and significance of this observation is very relevant to the ICLR community -- though the method is simple, to our knowledge no prior work has observed that such a simple strategy is so effective, as evidenced by the significant number of much more complex methods that have been proposed that attempt to learn or infer the reward function in such cases.
>
> Moreover, we have now conducted extensive **diagnostic studies** and **theoretically analyzed** UDS and CUDS to understand why these methods work well both in theory and practice, and added these to the revised paper in **Appendix F and G** respectively. Finally, we added experiments to evaluate UDS and CUDS in the more general dense reward setting where UDS and CUDS still achieve superior performance over prior approaches. All of these changes are in $\textcolor{red}{\text{red}}$. We briefly discuss these findings below, and **please let us know if this has addressed the main issues raised in your review.**
>
> We address individual comments below:
>
> >**The proposed data sharing methods (CUDS, UDS) will be beneficial even suppress the oracle. However, there is not a clear statement or discussion about the intuition when and why it can achieve this?**
>
> To understand the intuition when and why CUDS and UDS work, we have now updated the paper to incorporate a theoretical analysis of both of these approaches in **Appendix F**, and performed diagnostic experiments on the tasks studied in this paper to actually validate these insights gained from the theoretical analysis in **Appendix G**. We discuss them briefly below.
>
> **Theoretical Analysis:** We show that CUDS and UDS enjoy policy improvement guarantees and are able to obtain strong policy improvement bounds when we have long-horizon tasks and a large number of relabeled data that can be used to increase the coverage of the offline data and decrease the sampling error. Our theoretical analysis uses a novel tool that allows us to provide stronger guarantees (by exploiting the fact that the relative order of actions matters most for policy improvement compared to the absolute value of the Q-function), than naively applying prior analyses of offline RL.
>
> We also discuss two cases in Appendix F.2.3 where UDS can perform better than No Sharing despite the reward signal being suppressed. Please let us know if this answers your question.
>
> **Empirical insights for why UDS and CUDS work:** Our theoretical analysis suggests that UDS will help the most on domains with limited data or narrow coverage or low data quality, even when they suppress the reward signal.
>
> To test this, we choose the door-open task with three different combinations of dataset size and data quality, namely 2k transitions with the expert-level performance, 2k transitions with medium-level performance and a medium-replay dataset with 152k transitions, which represent the cases of high-quality data with limited sample size and narrow coverage, low-quality data with limited sample size and narrow coverage, and low-quality data with sufficient sample size and broad coverage respectively. We share the same data from the other three tasks, door-close, drawer-open and drawer-close as in Table 1. As shown in the table below, both UDS and CUDS are able to outperform No Sharing in the three settings except that UDS fails to achieve non-zero success rate in the medium-replay setting with broader coverage. Such results suggest that increasing the coverage of the offline data as suggested by our theory does lead to performance boost regardless of data quality and sample size. It’s worth noting that the UDS is unable to help in the setting where the data coverage is already broad and the data quality is low but CUDS mitigates the issue via a better data sharing strategy.
>
> | Environment | Dataset type / size | CUDS (ours)      | UDS (ours)           | No Sharing |
> |-------------|-----------------|-----------------|----------------|------------------|
> |  | expert (narrow) / 2k transitions | 67.6%   | 58.8% | 31.3%    |
> | Meta-World door-open  | medium (narrow) / 2k transitions | 67.3%   | 74.2% | 27.6%    |
> |  | medium-replay (broad) / 152k transitions | 30.0%  | 0.0% | 14.8%    |

---

> > ### Author Response · Authors · 2021-11-15
> > **Author Response (Part 2): Experiments on dense-reward settings, Clarifications on intuition of the methods**
> >
> > >**It is still confusing whether these data-sharing methods can be utilized in other application scenarios...there is an assumption in the targeted MDP is that the reward is binary...more general scenarios.**
> >
> > We actually also showed that UDS and CUDS can be applied to more general settings **with dense rewards** in the website linked in the appendix. We have now moved this to the paper in **Appendix D** and added a discussion of these experiments. We evaluate UDS and CUDS in the multi-task walker environment with three tasks, running forward, running backward and jumping as used in prior work [1]. In such a dense reward setting, we relabel the rewards of data shared from other tasks as the minimum reward in the target offline dataset. As shown in the table below, UDS and CUDS are still able to outperform No Sharing and achieve competitive performance compared to CDS and Sharing All with the oracle reward functions. This suggests that UDS and CUDS are able to be utilized in more general applications and can work without assuming binary rewards. We have moved this table to the revised version of the paper in Table 7 in Appendix D.
> >
> > | Environment |  Tasks / Dataset type | CUDS (ours)      | UDS (ours)           | No Sharing  | CDS (oracle) | Sharing All (oracle)|
> > |-------------|-----------------|-----------------|----------------|------------------|----------------|------------------|
> > | | run forward / medium-replay | 880.1$\pm$108.8 | 665.0$\pm$84.9  | 590.1$\pm$48.6 | **1057.9**$\pm$121.6 | 701.4$\pm$47.0|
> > |walker2d | run backward / medium | 717.8$\pm$78.3 | 689.3$\pm$16.3 | 614.7$\pm$87.3 | 564.8$\pm$47.7 | **756.7**$\pm$76.7|
> > | | jump / expert | 1487.7$\pm$177.6 | 1036.0$\pm$247.1  | **1575.2**$\pm$70.9 | 1418.2$\pm$138.4 | 885.1$\pm$152.9 |
> > | |  **average** |  **1028.6**$\pm$76.8 |   796.7$\pm$106.3  |  926.6$\pm$37.7 |  1013.6$\pm$71.5 |  781.0$\pm$100.8|
> >
> >
> > Our theoretical analysis in **Appendix F** also only assumes that the reward function can take on non-binary values in the range [0, 1]. We believe that this assumption of lying in [0, 1] is not necessary and can be easily relaxed to any general range of values for the reward function.
> >
> > ___
> >
> > > **The intuition of the great performance is also straightforward.**
> >
> > While at first it might appear that CUDS and UDS work well because they can estimate the transition kernel better and certainly it is a part of the reason why UDS and CUDS work, however we would argue that this is not quite this straightforward: in our experiments (Table 4), we compared with a prior method ACL (Yang et al. 2021) that enforces an auxiliary objective to **explicitly** learn the dynamics of the MDP using unlabeled data, and found it to be worse than UDS. We also added an experiment in Appendix E (Table 8) that compares our method to a prior model-based offline RL method COMBO (Yu et al. 2021) on the Meta-World domain in the table below where both UDS and CUDS outperformed COMBO. If UDS were to work solely due to a better estimation of the transition kernel, prior methods ACL and COMBO  should have worked much better than (or at least comparable to) UDS, since they directly learn the transition kernel, but the results suggest otherwise. This suggests that there are other factors at play that dictate the efficacy of UDS. Our theoretical results (Appendix F) highlight these factors (reward bias, sampling error, performance of the behavior policy) and the tradeoffs between them, and our empirical analysis that we now add in Appendix G verifies the effects of these factors.
> >
> > | Environment | CUDS (ours)      | UDS (ours)           | COMBO |
> > |-------------|-----------------|----------------|------------------|
> > | Meta-World (average task)  | 71.2% $\pm$ 11.3%   | 56.4%$\pm$12.8% | 25.6%$\pm$6.2%    |
> >
> > ___
> >
> > > **Minor comments and Remark 4.2.**
> >
> > Thank you for pointing out these issues. We have highlighted or fixed them in the revised version of the paper. We have also attempted to clarify Remark 4.2. Please let us know if it is still unclear and we are happy to explain/clarify it further.

---

> > > ### Comment · Reviewer_SFTg · 2021-11-21
> > > **Response to the author**
> > >
> > > Thanks so much for all the replies and answers.
> > > 1) I agree with the author that  CUDS/UDS still have other advantages except for a better estimation of the transition kernel. However, I can't see enough improvement beyond the most related work CDS, since the difference is only releasing the assumption of oracle reward. Although experimental results show CUDS/UDS have competitive results compared to CDS and the author also adds more theoretical results in Appendix F, it is still not very clear when and why CUDS/UDS can perform well intuitively.
> > > 2) Although CUDS/UDS outperforms other baselines such as COMBO, it may lead to the sharing strategy which already been proposed in the paper of CDS.
> > >
> > > So, unfortunately, I still think it is too similar to the previous work. The author may consider polishing the writing and adding more intuition and discussions about why no reward sharing can lead to good results.

---

> > > > ### Author Response · Authors · 2021-11-22
> > > > **Author response: Added intuition and more diagnostics for when UDS will work**
> > > >
> > > > Thank you for your reply! While we agree that CUDS is related to the prior paper (CDS), note that CDS cannot simply be used in settings where the reward function is unknown. Additionally, while the modification added by UDS is so simple (just label unlabeled data with the minimum possible reward), the fact that this simple UDS scheme works so well is notable and surprising since prior work [1, 2] has attempted to suggest significantly more complex solutions to deal with unlabeled data. Therefore, we believe that the good performance of UDS is of interest to the community as it is a simple method that outperforms very complex prior approaches, making it at least a simple, but strong baseline to compare to.
> > > >
> > > > As we discuss below, we have performed some more diagnostic experiments to identify when UDS will work (**Appendix G1 and G2**) and summarized these into intuitions (**Table 11, Appendix G.3**).
> > > >
> > > >
> > > > ___
> > > >
> > > >
> > > > > it is still not very clear when and why CUDS/UDS can perform well intuitively… adding more intuition and discussions about why no reward sharing can lead to good results.
> > > >
> > > > We would like to point the reviewer to an empirical analysis that indicates when and why UDS and CUDS work in practice from our previous response, in **Table 9** in **Appendix G** of the paper. Additionally, we have now performed a more detailed diagnostic study on the Hopper environment with dense rewards from D4RL (Fu et al. 2020) to understand and derive the intuitions behind UDS and these numbers are presented in **Table 10** in **Appendix G** in the revised paper (we also present this table at the very end of this response).
> > > >
> > > > We have now also added a table summarizing the takeaways from the empirical analysis (where **L** denotes the characteristics of labeled data, and **U** denotes the characteristics of unlabeled data), which enables us to tell when UDS and CUDS will work:
> > > >
> > > >
> > > > | Scenarios      | UDS (ours)           | Intuition |
> > > > |-------------|-----------------|-----------------|
> > > >  |**L**: limited + high-quality + narrow, **U**: abundant + low-quality + broad | **helps** | increase coverage|
> > > >   |**L**: limited + high-quality + narrow, **U**: abundant + medium-quality + narrow | **helps** | more negatives|
> > > >   |**L**: limited + medium-quality + narrow, **U**: abundant + low-quality + broad | **may not help** | reward bias outweighs high coverage|
> > > >   | **L**: abundant + medium-quality + broad, **U**: abundant + medium-quality + broad | **may not help** | reward bias outweighs high coverage|
> > > >  | **L**: limited + medium-quality + narrow, **U**: abundant + high-quality + narrow | **helps** | increase data quality |
> > > >  |  **L**: limited + low-quality + broad, **U**: abundant + medium-quality + narrow | **helps** | increase data quality |
> > > >  | **L**: limited + low-quality + broad, **U**: abundant + high-quality + narrow | **helps** | increase data quality|
> > > >
> > > > We have added this summarizing table in **Appendix G.3**.
> > > >
> > > > ___
> > > >
> > > >
> > > > > Although CUDS/UDS outperforms other baselines such as COMBO, it may lead to the sharing strategy which already been proposed in the paper of CDS.
> > > >
> > > > We agree that the reason that CUDS outperforms COMBO might be partly related to the data sharing strategy proposed in CDS, but we would like to clarify that UDS also outperforms COMBO without using any advanced sharing strategy, which suggests that our method does have its own benefit beyond CDS as indicated in our theoretical analysis.
> > > >
> > > > ___
> > > >
> > > > **Hopper diagnostic study (detailed performance)**
> > > >
> > > > | Environment | Labeled dataset type / size | Unlabeled dataset type / size      | UDS (ours)           | No Sharing |
> > > > |-------------|-----------------|-----------------|----------------|------------------|
> > > > |  | expert / 10k transitions | random / 1M transitions   | **90.8** | 77.1    |
> > > > |  | expert / 10k transitions | medium / 1M transitions   | **87.6** | 77.1    |
> > > > | D4RL hopper  | medium / 10k transitions | random / 1M transitions    | 9.8 | **28.7**    |
> > > > |  | medium / 10k transitions | expert / 1M transitions    | **106.1** | 28.7   |
> > > > |  | random / 10k transitions | medium / 1M transitions   | **51.9** | 9.6    |
> > > > |  | random / 10k transitions | expert / 1M transitions   | **97.0** | 9.6    |
> > > >
> > > > [1] Eysenbach, Benjamin, Sergey Levine, and Ruslan Salakhutdinov. "Replacing Rewards with Examples: Example-Based Policy Search via Recursive Classification." arXiv preprint arXiv:2103.12656 (2021).
> > > >
> > > > [2] Fu, Justin, et al. "Variational inverse control with events: A general framework for data-driven reward definition." arXiv preprint arXiv:1805.11686 (2018).

---

> > > > > ### Comment · Reviewer_SFTg · 2021-11-29
> > > > > **Response to author**
> > > > >
> > > > > Thanks for all your answer to my questions and concerns. The answers have addressed almost all my concerns. Although I partly agree with the author that CUDS and UDS outperform reasonable baselines and have their own benefits, it is still not convincing to me that no sharing reward leads to better results or almost the same results compared to sharing reward oracle. I will raise my score to 5.

---

> > > > > > ### Author Response · Authors · 2021-11-29
> > > > > > **Thank you for the response and raising the score; Clarifications and experiments for the remaining concern**
> > > > > >
> > > > > > Thank you for your reply and raising the score!
> > > > > >
> > > > > > On an average of all our experiments, we find that CUDS and UDS only perform comparable to CDS and Sharing All with true rewards, and do not outperform these significantly. That said, we agree with the reviewer that this result is surprising. We never claimed that CUDS and UDS should outperform CDS and Sharing All in the paper, and will modify the revised version to clearly indicate that this is a surprising finding. Our goal was to only show that CUDS and UDS outperform No Sharing, and our experiments and theory characterizes when this is possible. We provide some intuition for why this happens below, but we believe that this does not undermine the primary claims of the paper, and only provides an interesting avenue for future work to study.
> > > > > >
> > > > > > **Intuition:**. We believe that in our experiments, the labeled dataset provides a sufficient learning signal for the underlying algorithm to identify high-reward regions (*positives*) and UDS and CUDS serve as a way to provide useful information of what the learned policy shouldn’t do (i.e., “negatives”). To see why this is true, note that the labeled dataset in our experiments is either a small number of expert trajectories [**Meta-World door open / drawer close, walker jump**] or a whole replay buffer until the data collecting policy reaches the medium-level performance [**Meta-World door close / drawer open, walker run forward, MT-Opt tasks**]. Therefore, the labeled dataset contains enough high-reward transitions for UDS and CUDS to achieve competitive performances compared to Sharing All and CDS. Note however that these are the exact domains used by the prior work, CDS and Sharing All and we utilize performance numbers from prior work as well.
> > > > > >
> > > > > > **Additional experiment to validate our intuition:** To further empirically validate the intuition, we added an experiment that uses the hardest manipulation task (blocked drawer 2) from prior work, COG ([1] Singh et al. 2020). We create various settings by masking out rewards in different percentages of the dataset and instead present this masked out subset of data as unlabeled data. As shown in the table below, UDS with relabeling 50% of the data and also relabeling 50% of the data with 50% success rate achieves slightly better results than Sharing All. However, when we zero out 50%  data with a success rate of 60%, the performance of UDS drops significantly. This suggests that UDS performs similarly to Sharing All under the condition that the labeled data should provide sufficient supervision signals. We will add this empirical result and the discussion to the revised version of the paper.
> > > > > >
> > > > > > | Environment | COG / Sharing All    | UDS (relabeling 50% task-specific data) | UDS + relabeling 50% task-specific data with success rate 0.5 | UDS + relabeling 50% task-specific data with success rate 0.6|
> > > > > > |-------------|-----------------|-----------------|----------------|------------------|
> > > > > > | COG blocked drawer 2 | 0.76   | 0.81 | 0.81    | 0.56 |
> > > > > >
> > > > > > **Conclusion:** We attempted to provide the intuition for why UDS and CUDS still have performance comparable to their oracle counterparts. We agree this is surprising, but we believe that this does not undermine the primary claims of the paper in any way and is an interesting avenue for future work.
> > > > > >
> > > > > > [1] ​​Singh, Avi, et al. "Cog: Connecting new skills to past experience with offline reinforcement learning." arXiv preprint arXiv:2010.14500 (2020).
> > > > > >
> > > > > > ___
> > > > > >
> > > > > > **Please let us know if this addresses your concern.**

---

> > > > ### Author Response · Authors · 2021-11-27
> > > > **A Gentle Reminder for Discussion**
> > > >
> > > > Dear Reviewer SFTg,
> > > >
> > > > We hope that you've had a chance to read our response to your comment below and the updated paper where we have added intuitions for when and why UDS/CUDS works.
> > > >
> > > > We would really appreciate a reply before the end of the discussion period (discussion ends in 3 days) as to whether we have addressed your concerns or if any additional concerns remain. We are happy to address any remaining concerns and are eagerly looking for your feedback on the revised paper.
> > > >
> > > > Thanks so much!

---

> ### Author Response · Authors · 2021-11-20
> **Request for Discussion**
>
> Dear Reviewer SFTg,
>
> We hope that you've had a chance to read our response and the revised paper. We would really appreciate a reply before the end of the discussion period on Nov 22 as to whether our response and clarifications have addressed the issues raised in your review, or whether there is anything else we can address. We would be happy to provide further revisions or experiments to address any remaining issues.

---

### Official Review · Reviewer_UDVV · 2021-11-01

**Correctness:** 3
**Technical Novelty And Significance:** 2
**Empirical Novelty And Significance:** 2
**Recommendation:** 5
**Confidence:** 3

**Details Of Ethics Concerns:**

No ethics concerns.

**Main Review:**

Pros:
- The problem addressed in the paper is indeed critical in offline RL. It has a reasonable motivation.

Issues:
- The biggest concern is the novelty. The proposed CUDS and UDS methods share similarities with one existing study (Yu et al. 2021a). Indeed the proposed method is more sophisticated than this previous paper; however I would expect the authors to emphasize that there exists a substantive difference between the two different works.

- In this work’s multi-task offline RL setting, the authors assume the dynamics to be the same across all tasks and use a single task-conditioned policy. It is okay to assume the tasks only differ in reward distributions but I would expect a clearer justification.

- How does the proposed method differ from learning environment dynamics with more data (in a maybe model-based offline RL setting)? More insights on this point would be appreciated.


**Summary Of The Paper:**

This paper proposed a method for multi-task offline reinforcement learning setting where it can assign shared and annotate the reward, where its major strategy is to label only useful transitions. Specifically, the proposed method followed the conservative learning setting with the hope of easing the distribution shift issue. The authors also study the behavior of their proposed methods with s set of arguments and experimental results.

**Summary Of The Review:**

The major issue of this paper is it seems to be an incremental study from existing works. I would expect the authors to address this issue and add more insights.

---

> ### Author Response · Authors · 2021-11-15
> **Author Response (Part 1 of 3): Summary of Changes, Differences from prior works**
>
> Thank you for your comments! We believe your feedback has helped us improve the paper. We have attempted to address all the concerns raised in the review by updating the paper (changes in $\textcolor{red}{\text{red}}$).
>
> To begin with, we note that while CUDS and UDS are indeed simple modifications to the existing CDS approach, we would like to emphasize that CUDS and UDS remove the assumption of having access to the oracle reward functions of each task, which is problematic in practice as we discuss below. Even though CUDS and UDS are with much weaker assumptions, they can enjoy good empirical results as shown in Table 1. Hence, we believe that the empirical novelty and significance of this observation is major and relevant to the ICLR community.
>
> Moreover, we have now conducted **several diagnostic studies** and theoretically analyzed UDS and CUDS to understand why these methods work well both in theory and practice and added these insights to the revised paper in **Appendix F** and **G** respectively.  We also added an empirical comparison between UDS and CUDS and a prior model-based offline RL method COMBO, where our method achieves better performance compared to COMBO. We briefly discuss these findings below, and please let us know if this has addressed the main issues raised in your review.
>
> ___
>
> > **However, I would expect the authors to emphasize that there exists a substantive difference between the two different works.**
>
> The main difference between CUDS/UDS and the prior work CDS is that CDS requires knowing the true reward value for each task for all transitions in the shared data, whereas UDS and CUDS do not need to know the value of the reward for any of the shared data. This is a major difference in assumptions. Removing such assumptions is important for practical offline multi-task RL since it can be prohibitively expensive to label all data points for all tasks. Concretely, CUDS and UDS are able to forgo $\mathcal{O}(N^2)$ reward labels in practice where N is the number of tasks.
>
> While from an algorithm design perspective this difference is small, as shown in Table 1, CUDS and UDS are able to outperform several sophisticated prior methods that try to learn the (unknown) reward function, such as VICE and RCE, which suggests that our method is simple yet effective in practice. Though the method is simple, to our knowledge no prior work has observed that such a simple strategy is so effective, as evidenced by the large number of much more complex methods that have been proposed that attempt to learn or infer the reward function in such cases. Simple but effective changes to existing algorithms that perform well in practice are quite desirable and of interest to the community since they are easy to reproduce and build on.

---

> > ### Author Response · Authors · 2021-11-15
> > **Author Response (Part 2 of 3): Added theoretical and empirical insights for why UDS and CUDS work**
> >
> > > **Add more insights of the method.**
> >
> > To understand the insights behind why CUDS and UDS work, we have now updated the paper to incorporate a theoretical analysis of both of these approaches in **Appendix F**, and performed diagnostic experiments on the tasks studied in this paper to actually validate these insights gained from the theoretical analysis in **Appendix G**. We discuss them briefly below.
> >
> > **Theoretical Analysis:** We show that CUDS and UDS enjoy policy improvement guarantees and are able to obtain strong policy improvement bounds when we have long-horizon tasks and a large number of relabeled data that can be used to increase the coverage of the offline data and decrease the sampling error. Our theoretical analysis uses a novel tool that allows us to provide stronger guarantees than naively applying prior analyses of offline RL.
> >
> > **Empirical validation and gaining insights:** Our theoretical analysis suggests that UDS will help the most on domains with limited data or narrow coverage or low data quality.
> >
> > To test this, we have added an experiment in **Appendix G**, which we also discuss here. We used the door-open task from the Meta-World domain with three different combinations of dataset size and data quality, namely 2k transitions with the expert-level performance, 2k transitions with medium-level performance, and a medium-replay dataset with 152k transitions, which represent the cases of high-quality data with limited sample size and narrow coverage, low-quality data with limited sample size and narrow coverage, and low-quality data with sufficient sample size and broad coverage respectively. We share the same data from the other three tasks, door-close, drawer-open, and drawer-close as in Table 1. As shown in the table below, both UDS and CUDS are able to outperform No Sharing in the three settings, suggesting that increasing the coverage of the offline data as suggested by our theory does lead to performance boost in wherever we have limited good-quality data (expert), limited low-quality data (medium) and abundant low-quality data (medium-replay). It’s worth noting that UDS and CUDS significantly outperform No Sharing in the limited expert and medium data setting whereas, in the medium-replay setting with broader coverage, CUDS outperforms No sharing but UDS fails to achieve a non-zero success rate. Such results suggest that UDS and CUDS can yield greater benefit when the target task data has extremely narrow coverage while the improvement is limited in settings where the data coverage is already broad.
> >
> > | Environment | Dataset type / size | CUDS (ours)      | UDS (ours)           | No Sharing |
> > |-------------|-----------------|-----------------|----------------|------------------|
> > |  | expert (narrow) / 2k transitions | 67.6%   | 58.8% | 31.3%    |
> > | Meta-World door-open  | medium (narrow) / 2k transitions | 67.3%   | 74.2% | 27.6%    |
> > |  | medium-replay (broad) / 152k transitions | 30.0%  | 0.0% | 14.8%    |

---

> > > ### Author Response · Authors · 2021-11-15
> > > **Author Response (Part 3 of 3): Differences from model-based offline RL, References**
> > >
> > > > **How does the proposed method differ from learning environment dynamics with more data (in a maybe model-based offline RL setting)?**
> > >
> > > To address this question, we added an experiment in **Table 8 in Appendix E** that compares CUDS and UDS to a recent, state-of-the-art model-based offline RL method COMBO [1] in the Meta-World domain. As shown in the results below, CUDS and UDS indeed outperform COMBO in the average task success rate. The intuition behind this is that COMBO is unable to learn an accurate dynamics model for tasks with limited data as in our Meta-World setting.
> > >
> > > | Environment | CUDS (ours)      | UDS (ours)           | COMBO |
> > > |-------------|-----------------|----------------|------------------|
> > > | Meta-World (average task)  | 71.2% $\pm$ 11.3%   | 56.4%$\pm$12.8% | 25.6%$\pm$6.2%    |
> > >
> > > The original submission also already included the comparison between our method and a recently proposed *model-based representation learning for offline RL* method (ACL) in Table 4, where CUDS and UDS outperform ACL.
> > >
> > > The intuition behind these results is as follows. CUDS and UDS use the data from other tasks directly for learning the Q-functions and policy and do not learn the model of the environment dynamics and the reward function like model-based offline RL methods or offline RL with model-based representations. Therefore, CUDS and UDS can alleviate the challenge of handling model errors and do not require learning a reward predictor, which may be erroneously optimistic, when the amount of target data is limited, causing the policy to deviate away from the data distribution. The results suggest that this intuition may be true.
> > >
> > > ___
> > >
> > > > **It is okay to assume the tasks only differ in reward distributions but I would expect a clearer justification.**
> > >
> > > We follow the assumption from prior works in data sharing for multi-task RL [1,2,3]. While the setting is not fully general, there are many practical scenarios where only the reward changes and the dynamics remain the same such as various object manipulation objectives [4], different goal navigation tasks [5], and distinct user preferences [6]. Therefore, we follow such a setting in our work. We have revised the paper to include this discussion.
> > >
> > > ___
> > >
> > >
> > > [1] Yu, Tianhe, et al. "Conservative data sharing for multi-task offline reinforcement learning." arXiv preprint arXiv:2109.08128 (2021).
> > >
> > > [2] Kalashnikov, Dmitry, et al. "MT-Opt: Continuous Multi-Task Robotic Reinforcement Learning at Scale." arXiv preprint arXiv:2104.08212 (2021).
> > >
> > > [3] Eysenbach, Benjamin, et al. "Rewriting history with inverse rl: Hindsight inference for policy improvement." arXiv preprint arXiv:2002.11089 (2020).
> > >
> > > [4] Annie Xie, Avi Singh, Sergey Levine, and Chelsea Finn. Few-shot goal inference for visuomotor learning and planning. In Conference on Robot Learning, pages 40–52. PMLR, 2018.
> > >
> > > [5] Justin Fu, Aviral Kumar, Ofir Nachum, George Tucker, and Sergey Levine. D4rl: Datasets for deep data-driven reinforcement learning, 2020.
> > >
> > > [6] Paul Christiano, Jan Leike, Tom B Brown, Miljan Martic, Shane Legg, and Dario Amodei. Deep reinforcement learning from human preferences. arXiv preprint arXiv:1706.03741, 2017.

---

> ### Author Response · Authors · 2021-11-20
> **Request for Discussion**
>
> Dear Reviewer UDVV,
>
> We hope that you've had a chance to read our response and the revised paper. We would really appreciate a reply before the end of the discussion period on Nov 22 as to whether our response and clarifications have addressed the issues raised in your review, or whether there is anything else we can address. We would be happy to provide further revisions or experiments to address any remaining issues.

---

> > ### Author Response · Authors · 2021-11-27
> > **A Gentle Reminder for Discussion**
> >
> > Dear Reviewer UDVV,
> >
> > We hope that you've had a chance to read our response below and the revised paper. We would really appreciate a reply before the end of the discussion period (discussion ends in 3 days) as to whether we have addressed your concerns or if any additional concerns remain. We are happy to address any remaining concerns and are eagerly looking for your feedback on the revised paper.
> >
> > Thanks so much!

---

### Official Review · Reviewer_YPSx · 2021-11-03

**Correctness:** 4
**Technical Novelty And Significance:** 2
**Empirical Novelty And Significance:** 2
**Recommendation:** 5
**Confidence:** 3

**Main Review:**

Strength:
Two remarks in Section 4 provide an interesting intuition about how to use unlabeled transitions depending on whether each transition can reach a labeled state-action pair.
This work has empirically shown that a simple constant reward (zero reward) is enough to achieve good performance in practice.

Weakness:
Considering extending CDS into multi-task RL without oracle reward function, the simplest direction is relabeling unlabeled transitions with minimum reward. But it is not clear whether the direction is valid. Therefore, the readers expect justification of using simple relabeling. However, CUDS is only supported empirically without any theoretical analysis.

Proposition 4.1 is proposed to prove the Q-value function is underestimated by UDS. However, it is not clear why such lower estimation is desirable in offline multi-task RL. Specifically, it seems too conservative way to estimate unlabeled rewards. For example, if we consider the $(s,a)$ is in both labeled and unlabeled data, then the reward of the state-action pair may be underestimated.
In addition, it is trivial since $\hat{r}(s,a)\le r(s,a)$ for all $(s,a)$ pairs.


**Summary Of The Paper:**

This paper proposes a new offline multi-task RL algorithm named conservative unsupervised data sharing (CUDS). Prior works on multi-task RL require the functional form of the reward functions or are limited to goal-conditioned setting. On the other hands, CUDS shares task-agnostic transitions via simply relabeling their rewards as a constant.
The authors first prove that $\hat{Q}_\text{UDS}\le \hat{Q}_\text{sharing All}$ for all state-action pairs. Furthermore, they argue that the CUDS can achieve better performance through weighting transitions.
Finally, CUDS outperforms or performs competitively with the previous works. Surprisingly, CUDS achieves competitive results compared to CDS, which uses true reward functions for relabeling


**Summary Of The Review:**

The remarks and empirical results are interesting. However, CUDS is a simple variant of CDS – the only difference is how to set the rewards. Therefore, I think CUDS needs additional meaningful theoretical analysis in addition to empirical results.

---

> ### Author Response · Authors · 2021-11-15
> **Author Response: Added Theoretical Analysis in Appendix F**
>
> Thank you for your comments! We believe this feedback has helped us improve the paper We have attempted to address all the concerns raised in the review and have uploaded a revised paper. In summary, we added a theoretical analysis of UDS and CUDS in **Appendix F** and have added experiments to understand why UDS and CUDS work well (**Appendix G**). All of these changes are in $\textcolor{red}{\text{red}}$.
>
> We briefly discuss these findings below. **Please let us know if this has addressed the main issues raised in your review.**
>
> ___
>
> > **CUDS is only supported empirically without any theoretical analysis... I think CUDS needs additional meaningful theoretical analysis in addition to empirical results**
>
> To resolve this concern, we have added a theoretical analysis of UDS and CUDS in Appendix F. **Proposition F.1** shows that UDS enjoys a policy improvement guarantee despite the fact that UDS introduces a term of reward bias due to relabeling zero rewards. Note that this analysis utilizes novel proof techniques compared to prior analyses, a direct application of which would just give rise to very loose bounds in our case (**see Appendix F.2.1**). We discuss cases where this guarantee is superior to the guarantee for No Sharing such as tasks with long horizons and in the case where the unlabeled data also consists of good transitions.
>
> We extend the policy improvement guarantee for UDS to CUDS in **Corollary F.1** and note that in addition to the benefit of reduced sampling error that UDS offers, especially in long-horizon tasks, CUDS also reduces suboptimality of the dataset after adding shared data by controlling distributional shift. Therefore, CUDS and UDS enjoy both meaningful theoretical guarantees and good empirical results.
>
> We would appreciate it if you can check these guarantees and tell us if these address your concerns. We are happy to clarify or extend any analyses here.
>
> ___
>
>
> > **Proposition 4.1 is proposed to prove the Q-value function is underestimated by UDS. If the state-action pair is in both labeled and unlabeled data it should be underestimated.**
>
> Per the reviewer’s suggestion, we have moved Proposition 4.1 to Appendix A since it is easy to prove. In addition, we have added new theoretical justifications for why UDS works in Appendix F. We agree that UDS is a conservative way to relabel rewards, however, our theoretical analysis of UDS in Appendix F shows that UDS can attain strong policy improvement guarantees, even when it is conservative. The intuition is that even though the rewards are conservative, policy selection simply aims to find the action with the highest Q-value, without considering the absolute magnitude of Q-values and may still find the optimal action with small values. We utilize this fact in Proposition F.1 and are able to still attain a good guarantee for policy improvement with UDS.

---

> > ### Comment · Reviewer_YPSx · 2021-11-29
> > **Response to the authors**
> >
> > I read the revised paper and agree that the theoretical analysis is improved.
> >
> > CUDS shows the good performance, but it is too similar to CDS. (e.g., overall algorithm, conservative sharing based on Q-value, etc.)
> >
> > In summary, I still believe that the work is very similar to CDS.

---

> > > ### Author Response · Authors · 2021-11-29
> > > **Thank you for your reply; Clarifications to the novelty concern**
> > >
> > > Thank you for your reply! While from an algorithm design perspective the difference between CUDS and CDS is small, note that CUDS and UDS remove the strong assumption of CDS and Sharing All that the reward functions of all tasks are known, which is impractical in practice. Moreover, as shown in Table 1, CUDS and UDS are able to outperform several sophisticated prior methods that try to learn the (unknown) reward function, such as VICE and RCE, which suggests that our method is simple yet effective in practice. Though the method is simple, to our knowledge no prior work has observed that such a simple strategy is so effective, as evidenced by the large number of much more complex methods that have been proposed that attempt to learn or infer the reward function in such cases. Simple but effective changes to existing algorithms that perform well in practice are quite desirable and of interest to the community since they are easy to reproduce and build on.

---

> ### Author Response · Authors · 2021-11-20
> **Request for Discussion**
>
> Dear Reviewer YPSx,
>
> We hope that you've had a chance to read our response and the revised paper. We would really appreciate a reply before the end of the discussion period on Nov 22 as to whether our response and clarifications have addressed the issues raised in your review, or whether there is anything else we can address. We would be happy to provide further revisions or experiments to address any remaining issues.

---

> > ### Author Response · Authors · 2021-11-27
> > **A Gentle Reminder for Discussion**
> >
> > Dear Reviewer YPSx,
> >
> > We hope that you've had a chance to read our response below and the revised paper. We would really appreciate a reply before the end of the discussion period (discussion ends in 3 days) as to whether we have addressed your concerns or if any additional concerns remain. We are happy to address any remaining concerns and are eagerly looking for your feedback on the revised paper.
> >
> > Thanks so much!

---

### Author Response · Authors · 2021-11-18
**Summary of Changes**

We thank the reviewers for their feedback. In this summary note, we would like to highlight our responses to the main concerns raised by the reviewers and the main experiments that we have added in the rebuttal period. We believe that we have addressed all of the concerns and also revised the paper accordingly (highlighted in $\textcolor{red}{\text{red}}$).

1. **[Reviewer YPSx, UDVV, SFTg]** We added a theoretical analysis of our method in **Appendix F**, which shows that CUDS and UDS enjoy safe policy improvement guarantees that are tighter than naively applying the analysis in prior works.

2. **[Reviewer UDVV, SFTg]** In addition to the theoretical analysis of our method, we included empirical analysis to validate our theory and provide insights for reasons that CUDS and UDS work in **Appendix G** and responses to Reviewers UDVV and SFTg.

3. **[Reviewer UDVV, SFTg]** Regarding the issue of minor changes on top of the prior method CDS, we would like to emphasize that CUDS and UDS remove the assumption of having access to the oracle reward functions of each task, which is problematic in practice. Moreover, UDS and CUDS enjoy good empirical results as shown in **Table 1** in the paper and also have theoretical guarantees as shown in our added analysis. Hence, we believe that the empirical novelty, theoretical soundness and significance of this observation is major and relevant to the ICLR community.

4. **[Reviewer SFTg]** To test if CUDS and UDS can be applied to more general settings without binary rewards, we added an experiment in a multi-task walker environment with dense rewards (**Table 7 in Appendix D**) where CUDS and UDS achieve superior results.

5. **[Reviewer UDVV]** Per the reviewer’s suggestion, we compared CUDS and UDS to the offline model-based RL method, COMBO (Yu et al. 2021) in **Table 8 in Appendix E** where CUDS and UDS outperform COMBO significantly.

We would appreciate it if the reviewers can please take a look at these changes and let us know if they have any other concerns or questions. Thank you!

---

### Author Response · Authors · 2021-11-23
**Summary of changes regarding the intuition about why our method works**

We would like to highlight that we have added an empirical analysis of the intuition why UDS/CUDS work in **Appendix G** on two multi-task environments and included a summarizing table on conditions where UDS works in practice in **Table 11 in Appendix G.3**. We believe that our new empirical analysis is able to address the concern on lack of intuition of our method raised by reviewers **SFTg** and **UDVV**. We would really appreciate it if the reviewers could go over our new empirical analysis and let us know if they have additional questions. Thank you!

---

### Decision · Program_Chairs · 2022-01-20

**Decision:**

Reject

**Comment:**

Although sharing data between tasks benefits multitask RL, this requires that rewards be relabeled across tasks. This paper shows that, for binary rewards, directly reusing data from other tasks with constant reward relabels is effective, and the paper develops a method around this idea that is highly effective.  The reviewers found that the idea and execution were impressive, that the paper was well written, and that the empirical analysis was convincing.

In response to concerns in the preliminary reviews about certain shortcomings in the empirical analysis and some lack of theoretical analysis, the authors provided substantial revisions to the paper. Due to some lack of reviewer response to the discussion, this meta-reviewer examined whether those revisions were sufficient to address the reviewers' concerns. The authors did a good job in providing the requested improvements and the analysis is stronger, but remaining similarities to existing methods (CDS) means that this paper still remains borderline. These same concerns were also shared by reviewers that continued to engage in discussion with the authors. To remedy this, the authors are encouraged to better and more substantially address differences with prior work in the writing and motivation throughout the entire paper. In addition, although space is a concern, it would be beneficial to integrate the high-level takeaways from the new analyses in the appendices into the main paper.